# A novel autophagy enhancer as a therapeutic agent against metabolic syndrome and diabetes

Hyejin Lim[1,2], Yu-Mi Lim[1], Kook Hwan Kim[1], Young Eui Jeon[3], Kihyoun Park[2], Jinyoung Kim[1], Hui-Yun Hwang[4], Dong Jin Lee[4], Haushabhau Pagire[5], Ho Jeong Kwon [4,6], Jin Hee Ahn[5] & Myung-Shik Lee [1,6]

Autophagy is a critical regulator of cellular homeostasis, dysregulation of which is associated with diverse diseases. Here we show therapeutic effects of a novel autophagy enhancer identified by high-throughput screening of a chemical library against metabolic syndrome. An autophagy enhancer increases LC3-I to LC3-II conversion without mTOR inhibition. MSL, an autophagy enhancer, activates calcineurin, and induces dephosphorylation/nuclear translocation of transcription factor EB (TFEB), a master regulator of lysosomal biogenesis and autophagy gene expression. MSL accelerates intracellular lipid clearance, which is reversed by lalistat 2 or *Tfeb* knockout. Its administration improves the metabolic profile of *ob/ob* mice and ameliorates inflammasome activation. A chemically modified MSL with increased microsomal stability improves the glucose profile not only of *ob/ob* mice but also of mice with diet-induced obesity. Our data indicate that our novel autophagy enhancer could be a new drug candidate for diabetes or metabolic syndrome with lipid overload.

[1] Severance Biomedical Science Institute, Seoul, Korea. [2] Department of Health Sciences and Technology, SAIHST, Sungkyunkwan University, Seoul, Korea. [3] Department of Oral Biology, Yonsei University College of Dentistry, Seoul, Korea. [4] Global Research Laboratory, Department of Biotechnology, College of Life Science and Biotechnology, Yonsei University, Seoul, Korea. [5] Department of Chemistry, Gwangju Institute of Science and Technology, Gwangju, Korea. [6] Department of Internal Medicine, Yonsei University College of Medicine, Seoul, Korea. Correspondence and requests for materials should be addressed to M.-S.L. (email: mslee0923@yuhs.ac)

Macro-autophagy is a cellular process involving lysosomal degradation of the cell's own material through formation of a new structure with double membranes (autophagosome) and its fusion to lysosome (autophagolysosome)[1]. The physiological roles of autophagy include quality control of organelles or cellular proteins and protection of nutrient balance[2]. Because autophagy is critical for the maintenance of cellular metabolic homeostasis, it plays a crucial role in the control of whole-body metabolism, dysregulation of which may participate in the development of metabolic disorders.

The in vivo role of autophagy in metabolic disorders has been widely studied using genetic models that showed diverse metabolic features[3]. For example, mice with knockout of Atg7, an essential autophagy gene in pancreatic β-cells producing insulin, showed structural and functional defects of pancreatic β-cells, resulting in glucose intolerance and susceptibility to diabetes in the presence of metabolic stress[4–6]. In contrast, autophagy knockout in skeletal muscle cells led to the induction of fibroblast growth factor 21 (FGF21) as a "mitokine" due to mitochondrial stress and resistance to diet-induced obesity and insulin resistance[7], in contrast to the expectation that autophagy deficiency associated with mitochondrial dysfunction in insulin target tissues would lead to insulin resistance. While these genetic models showed diverse metabolic phenotypes depending on the location and severity of autophagy deficiency, systemic autophagy insufficiency of physiologically relevant degree rather than tissue-specific knockout compromised adaptation to metabolic stress and facilitated progression from obesity to diabetes[8]. Furthermore, overexpression of Atg5, another essential autophagy gene, improved the metabolic profile of aged mice[9]. These results suggest that systemically enhanced autophagic activity may have beneficial effects on body metabolism during metabolic stress.

Since autophagy is involved in various biological processes and diseases, searches for autophagy modulators have been conducted to develop novel compounds with therapeutic effects against neurodegeneration, cancer or aging[10–13]. We screened a chemical library using a luciferase-based high-throughput assay of autophagic flux[14], rather than autophagy level, and identified novel small-molecule autophagy enhancers that can improve the metabolic profile through upregulation of lysosomal function.

## Results

**Screening of autophagy enhancer small molecules**. To screen autophagy enhancers, we stably transfected HepG2 cells with wild-type (WT) pRLuc(C124A)-LC3 or a mutant with G120A substitution that is resistant to proteolytic cleavage and inhibits LC3-II formation [pRLuc(C124A)-LC3(G120A)][14]. We treated transfectants with a chemical library comprising 7520 compounds (Korea Chemical Bank) for 24 h and selected 35 chemicals that reduced normalized the wild/mutant luciferase ratio to <0.6 at 50 μM concentration, due to autophagic degradation of Renilla luciferase without cytotoxic activity (viability of ≥80% determined by the 3-(4,5-dimethylthiazol-2-yl)-2,5-diphenyltetrazolium bromide (MTT) assay) (Supplementary Fig. 1a–c). A normalized wild/mutant luciferase ratio of 0.6 was chosen since 250 nM rapamycin, a positive control, reduced the ratio to 0.6. To confirm enhanced autophagic activity, we conducted Western blot analysis. Sixteen of 35 chemicals increased LC3-I to LC3-II conversion in the presence of bafilomycin A1 (Supplementary Fig. 1a, d), indicating that they are authentic autophagy enhancers. We next conducted Western blot analysis using anti-phospho-S6K1 and -phospho-mTOR antibodies to eliminate mTORC1 inhibitors that may exert deleterious effects on the metabolic profile and pancreatic β-cell function[15,16], and identified seven chemicals that did not inhibit mTORC1 (Supplementary Fig. 1e). Among them,

three chemicals (#6, #9 and # 30) improved the glucose profile of ob/ob mice after in vivo administration for 8 weeks in our preliminary experiments (Supplementary Fig. 2). We then selected one chemical (#9, hereafter called "MSL") for further in-depth experiments (Fig. 1a).

**Calcineurin activation by MSL enhanced autophagic flux**. We investigated the molecular mechanism of mTORC1-independent autophagy activation by MSL. Confocal microscopy after mRFP-GFP-LC3 transfection showed that MSL treatment induced the formation of red puncta representing the autophagolysosome, suggesting autophagy progression to the lysosomal step[17] (Fig. 1b). When we examined the lysosomal steps of autophagy more closely employing acridine orange (AO) staining, the number of acidic vesicles with red fluorescence was significantly increased by MSL treatment for 24 h (Fig. 1c), suggesting increased lysosomal content[18]. We therefore studied transcription factor EB (TFEB), a master regulator of lysosome biogenesis and autophagy gene expression[19]. Confocal microscopy showed TFEB nuclear translocation in >80% of cells treated with 50 μM MSL for 2 h (Fig. 1d). TFEB nuclear translocation was dose-dependent at MSL concentrations between 1–100 μM (Supplementary Fig. 3a). Western blot analysis showed increased TFEB mobility in cells treated with MSL, suggesting TFEB dephosphorylation[20] (Fig. 1e). Indeed, Western blot analysis using anti-phospho-S142-TFEB antibody confirmed reduced TFEB phosphorylation at S142, an important site of TFEB phosphorylation[19], by MSL treatment for 4 h (Fig. 1e). Furthermore, transfection of cells with a phosphomimetic Tfeb mutant [Tfeb(S142D)] markedly reduced nuclear translocation of TFEB by MSL treatment (Fig. 1f), suggesting a crucial role of S142 in TFEB localization by MSL treatment. To corroborate the role of TFEB nuclear translocation in autophagy activation, we treated CRISPR/Cas9 Tfeb knockout HeLa cells[21] with MSL. LC3-I to LC3-II conversion after 4 h of treatment was markedly reduced in Tfeb knockout cells (Fig. 1g), confirming an essential role of TFEB in autophagy activation by MSL.

We investigated the mechanism of MSL-induced TFEB dephosphorylation that is unrelated to the inhibition of mTORC1, a well-known inducer of TFEB phosphorylation[22]. Specifically, we studied the lysosomal $Ca^{2+}$-calcineurin pathway that induces TFEB dephosphorylation[23]. To determine whether MSL modulates lysosomal $Ca^{2+}$ release, HeLa cells were transfected with GCaMP3-ML1 encoding a lysosome-specific $Ca^{2+}$ probe[23]. MSL did not induce lysosomal $Ca^{2+}$ release, whereas a lysosomotropic agent (Gly-Phe β-naphthylamide, GPN) or ionomycin did (Fig. 2a). We therefore examined whether MSL increases calcineurin activity without inducing lysosomal $Ca^{2+}$ efflux. Calcineurin phosphatase activity was indeed significantly increased by MSL (Fig. 2b). The increase of calcineurin activity was dose-dependent at MSL concentrations between 1 and 100 μM (Supplementary Fig. 3b). Furthermore, MSL-induced calcineurin activation was suppressed by a combination of cyclosporin A (CsA) + FK506, calcineurin inhibitors (Fig. 2b), indicating that MSL increases calcineurin activity. Pretreatment with CsA + FK506 combination also abrogated TFEB translocation after MSL treatment for 4 h (Fig. 2c), supporting the role of calcineurin activation in TFEB translocation by MSL. To further confirm the role of calcineurin in MSL-induced TFEB translocation, we transfected Tfeb-GFP-transfectants with a dominant-negative mutant of catalytic calcineurin A subunit (HA-ΔCnA-H151Q)[24]. MSL-induced nuclear translocation of TFEB was markedly reduced in transfected cells (Fig. 2d), suggesting that calcineurin activation is crucial in nuclear translocation of TFEB by MSL. In contrast,

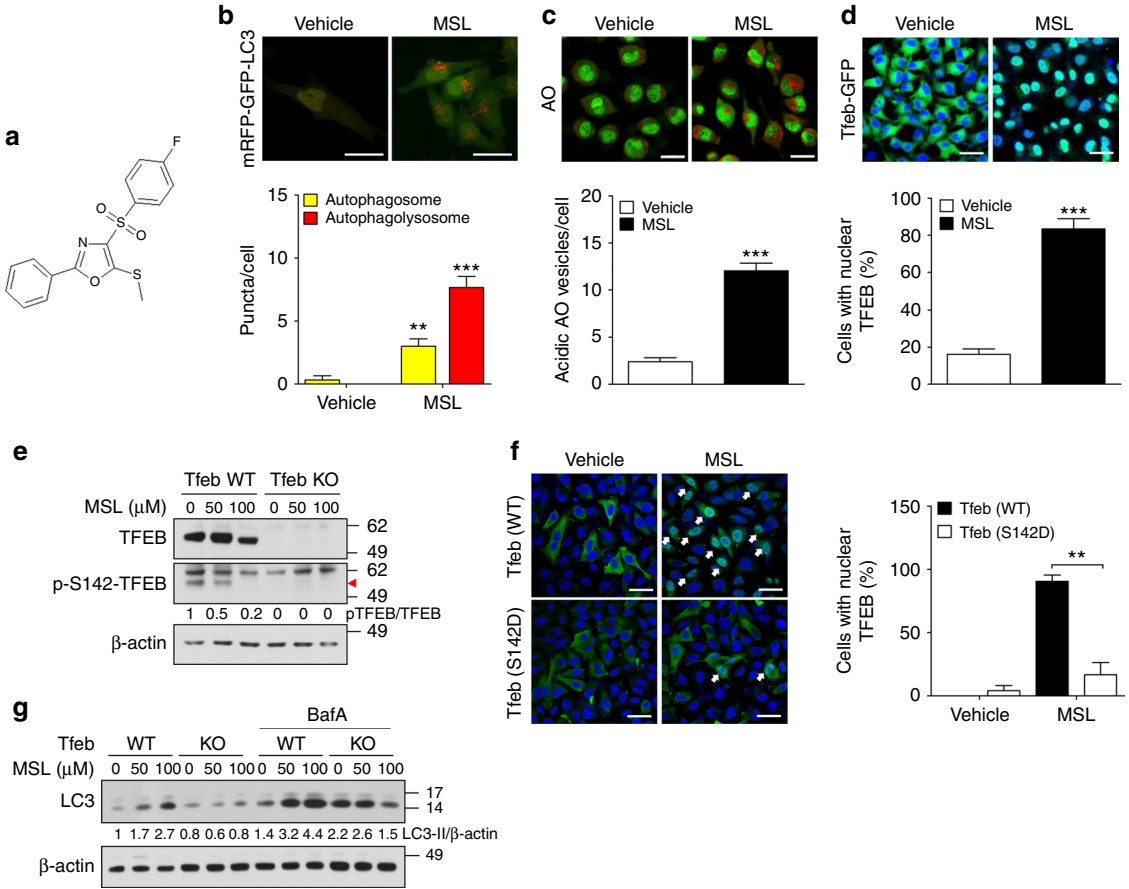

**Fig. 1** Identification of an autophagy enhancer small-molecule (MSL) inducing nuclear translocation of TFEB. **a** Chemical structure of autophagy enhancer, MSL [4-(4-fluorophenyl)sulfonyl-5-methylthio-2-phenyloxazole]. **b** HeLa cells transfected with *mRFP-GFP-LC3* were treated with MSL, and confocal microscopy was performed. Red puncta represents autophagolysosome (upper). The numbers of yellow and red punctae representing autophagosomes and autophagolysosomes, respectively, were counted ($t = 4.9$, df = 6 for autophagosome; $t = 10.3$, df = 6 for autophagolysosome) (lower). **c** MSL-treated cells were stained with AO, and confocal microscopy was performed (upper). The number of acidic AO vesicles was counted ($t = 10.3$, df = 4) (lower). **d** *Tfeb-GFP*-transfectants were treated with MSL, and then subjected to confocal microscopy to examine nuclear translocation of TFEB (upper). The number of cells with nuclear TFEB was counted ($t = 10.9$, df = 4) (lower). **e** WT and *Tfeb* knockout HeLa cells were treated with MSL, and cell extract was subjected to Western blot analysis using the indicated antibodies. Red arrow indicates phospho-S142-TFEB band. Numbers below phospho-S142-TFEB immunoblot bands indicate fold changes normalized to total TFEB bands. **f** HeLa cells transfected with 3x*FLAG-Tfeb*(WT) or -*Tfeb*(S142D) mutant were treated with MSL. Confocal microscopy was conducted after immunostaining with anti-FLAG antibody. Arrows indicate cells with nuclear translocation of TFEB (left panel). The number of cells with nuclear TFEB among FLAG⁺ cells was counted ($F = 51.7$, df treatment = 3, df residual = 6) (right). **g** WT and *Tfeb* knockout HeLa cells were treated with MSL in the presence or absence of bafilomycin A1 (BafA), and cell extract was subjected to Western blot analysis using the indicated antibodies. Numbers below LC3 immunoblot bands indicate fold changes of LC3-II normalized to β-actin bands. All data in this figure are the means ± s.e.m. from ≥3 independent experiments performed in triplicate (scale bar, 20 μm). **P < 0.01 and ***P < 0.001 by one-way ANOVA with Tukey's post-hoc test (**f**) and two-tailed Student's *t*-test (**b**–**d**)

transfection of constitutively active calcineurin mutant (*HA-ΔCnA*) together with regulatory calcineurin B subunit (*CnB*)[24] induced TFEB nuclear translocation without MSL treatment (Fig. 2d). Transfection of *HA-ΔCaN* and *CnB* also increased LC3-I to LC3-II conversion in the presence of bafilomycin A1, indicating enhanced autophagic flux (Supplementary Fig. 3c). To investigate the mechanism of calcineurin activation by MSL, we studied the physical interaction between MSL and calcineurin A employing a drug affinity responsive target stability (DARTS) assay[25]. When cell lysate was treated with pronase, calcineurin A stability was markedly decreased. Degradation by pronase was abrogated by pretreatment with a saturating dose of MSL (1 mM), suggesting that MSL directly binds to calcineurin A and stabilizes calcineurin A (Fig. 2e). Nonsaturating dose of MSL (100 μM) did not protect calcineurin A from degradation by pronase (Supplementary Fig. 4), probably because maximal protection of the target protein from proteolysis may require saturation of the

protein with ligand[25,26] and treatment of highly concentrated cell lysate with nonsaturating dose would not be enough for full protection of target proteins in DARTS assay.

**Accelerated clearance of intracellular lipid by MSL**. We next studied whether MSL can improve the cell's ability to handle metabolic stress through autophagy. When HeLa cells were loaded with palmitic acid (PA) + oleic acid (OA) combination for 24 h and then treated with MSL, BODIPY⁺ lipid accumulation was markedly reduced after 16 h of treatment (Fig. 3a), showing increased lipid clearance by MSL. We confirmed the expression of ATGL and HSL in HeLa cells (data not shown). BODIPY colocalized with transfected mRFP-LC3 or LAMP1 in cells treated with MSL for 1 h (Fig. 3b), suggesting direct interaction between lipid and autophagolysosome or occurrence of lipophagy. After 24 h of treatment, colocalization between BODIPY and LAMP1 was virtually absent, except a few droplets (Supplementary

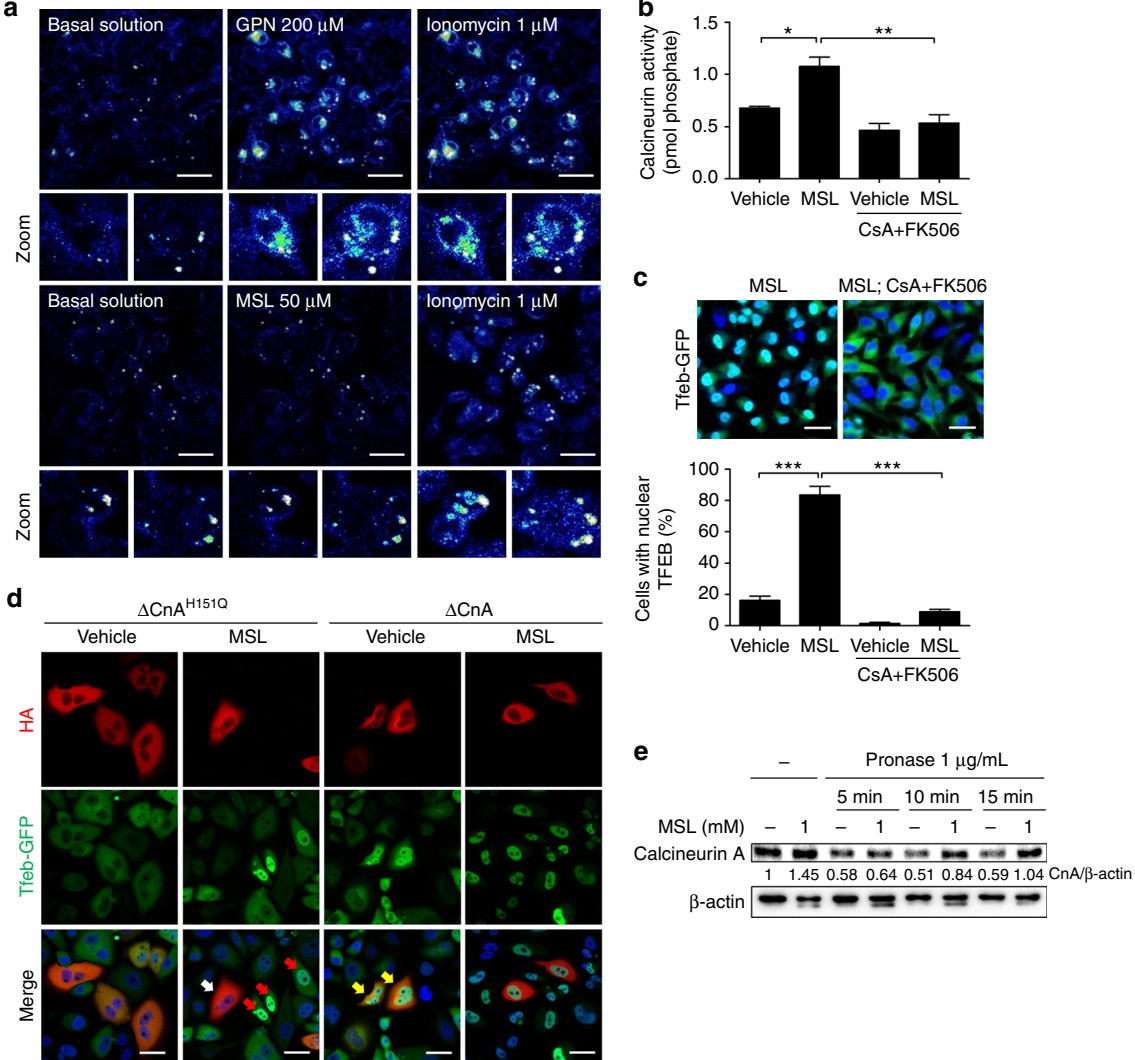

**Fig. 2** MSL-induced calcineurin activation increases autophagic flux. **a** HeLa cells transfected with *GCaMP3-ML1* encoding a lysosome-specific $Ca^{2+}$ probe were treated with GPN, ionomycin or MSL. Lysosomal $Ca^{2+}$ release was visualized by confocal microscopy. **b** Cells were treated with 50 μM MSL with or without pretreatment with CsA + FK506 combination ($F = 13.4$, df treatment = 3, df residual = 6). **c** *Tfeb-GFP*-transfectants were treated with MSL with or without pretreatment with CsA + FK506 combination, and then subjected to confocal microscopy (upper). The number of cells with nuclear TFEB was counted ($F = 133.5$, df treatment = 3, df residual = 11) (lower). **d** *Tfeb-GFP*-transfectants were transfected with *HA-ΔCnA*-H151Q or -*ΔCnA* construct, and analyzed by confocal microscopy after treatment with MSL for 4 h and immunostaining with anti-HA antibody (white arrow, *HA-ΔCnA*-H151Q-transfected cells showing no TFEB translocation by MSL; red arrows, *HA-ΔCnA*-H151Q-untransfected cells showing TFEB translocation by MSL; yellow arrows, *HA-ΔCnA*-transfected cells showing TFEB translocation without MSL). **e** Cell lysate was treated with pronase with or without MSL pretreatment, and subjected to Western blot analysis using anti-calcineurin A antibody. Numbers below immunoblot bands indicate fold changes normalized to β-actin bands (scale bar, 20 μm). All data in this figure are the means ± s.e.m. from ≥3 independent experiments performed in triplicate. *$P < 0.05$, **$P < 0.01$ and ***$P < 0.001$ by one-way ANOVA with Tukey's post-hoc test

Fig. 5a). The decrease of lipid content by MSL treatment was largely reversed by orlistat, a nonspecific lipase inhibitor, or lalistat 2 (kindly provided by Paul Helquist, University of Notre Dame, Notre Dame, IN), a specific lysosomal lipase inhibitor (Fig. 3a and Supplementary Fig. 5b), suggesting the role of lysosomal lipolysis in MSL-induced lipid clearance. Lipid clearance by MSL was also reduced by calcineurin inhibitors or bafilomycin A1 and in *Tfeb* or *Atg7* knockout cells (Fig. 3c, d and Supplementary Fig. 5b, c), demonstrating the role of calcineurin-mediated TFEB nuclear translocation and autophagolysosomal activity in this process. While MSL expedited lipid clearance after loading of PA + OA, MSL did not inhibit Hepa1c1c7 cell death by treatment with PA alone without OA, a condition that does not induce lipid droplet (LD) formation[27] (Supplementary

Fig. 5d), suggesting that MSL acts by inducing degradation of LD but does not directly inhibit PA-induced lipotoxicity. OA protected PA-induced cell death (Supplementary Fig. 5d), as previously reported[27].

NACHT, LRR, and PYD domains-containing protein 3 (NLRP3) inflammasome activation by lipid is a cause of metabolic inflammation and insulin resistance in obesity[28]. Since inflammasome activation by lipid is regulated by autophagy[29], we studied whether MSL can modulate inflammasome activation. When the effect of MSL on inflammasome activation by PA + lipopolysaccharide (LPS) combination[30] was studied, IL-1β release from macrophages was significantly attenuated (Supplementary Fig. 6a). Furthermore, Western blot analysis showed that IL-1β maturation, a marker of inflammasome activation, was

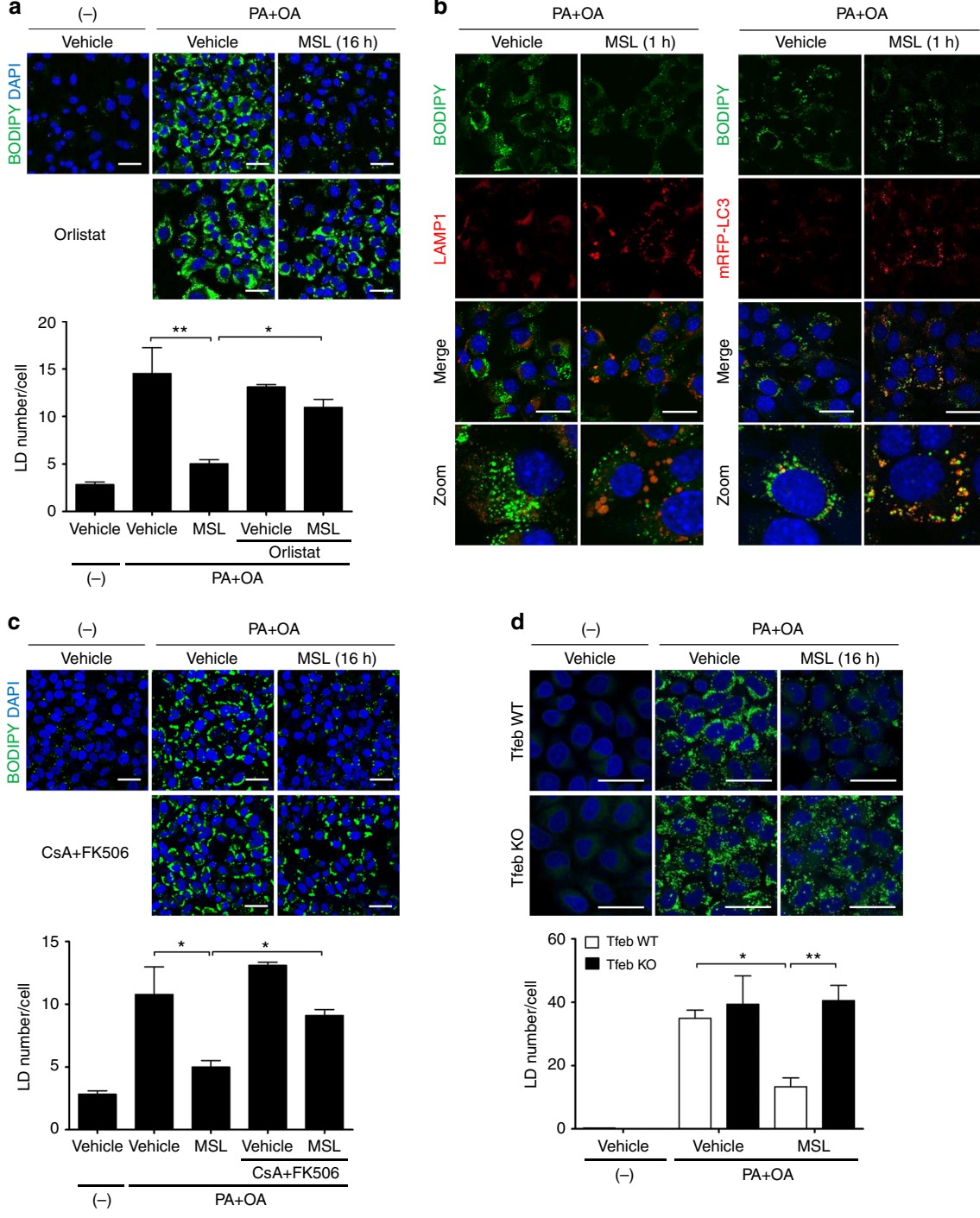

**Fig. 3** Decreased intracellular lipid after treatment with autophagy enhancer. **a** HeLa cells incubated in 2% fatty acid-free BSA-DMEM containing PA (400 μM) and OA (800 μM) for 24 h were treated with MSL in the presence or absence of orlistat. Control cells indicated as (-) were incubated in solvent alone (2% fatty acid-free BSA-DMEM). After staining with BODIPY493/503, cells were subjected to confocal microscopy to determine the number of lipid droplets (LDs) ($F = 14.3$, df treatment = 4, df residual = 8). **b** HeLa cells loaded with PA + OA combination as in **a** were treated with MSL. After immunostaining with anti-LAMP1 antibody and BODIPY493/503 staining, confocal microscopy was conducted (left panel). *mRFP-LC3*-transfected HeLa cells were loaded with PA + OA combination as in **a** and then treated with MSL. After staining with BODIPY493/503, confocal microscopy was conducted (right panel). **c** HeLa cells were treated and examined as in **a** after pretreatment with CsA + FK506 combination ($F = 15.9$, df treatment = 4, df residual = 8). **d** WT or *Tfeb* knockout HeLa cells were treated and examined as in **a** ($F = 21.3$, df treatment = 5, df residual = 10). All data in this figure are the means ± s.e.m. from ≥3 independent experiments performed in triplicate (scale bar, 20 μm). *$P < 0.05$ and **$P < 0.01$ by one-way ANOVA with Tukey's post-hoc test

reduced by MSL, supporting attenuated lipid-induced inflammasome activation by MSL (Supplementary Fig. 6b). Since autophagy controls inflammasome activation by regulating turnover of dysfunctional mitochondria[31], we measured mitochondrial reactive oxygen species (ROS). MSL significantly suppressed mitochondrial ROS production by treatment with PA + LPS combination for 24 h (Supplementary Fig. 6c), indicating attenuated mitochondrial dysfunction by MSL. Other markers of mitochondrial dysfunction such as decreased mitochondrial potential and suppressed ATP-coupled mitochondrial oxygen consumption after treatment with PA + LPS combination were also ameliorated by MSL (Supplementary Fig. 6d, e). During the study of inflammasome activation, we observed that MSL reduced not only mature IL-1β protein level but also pro-IL-1β protein level (Supplementary Fig. 6b), which could be due to autophagy-independent mechanisms. Thus, we determined the mRNA level of cytokines such as pro-IL-1β, TNFα, and IL-6 that is not directly affected by inflammasome activation. Indeed, mRNA expression of these cytokines after LPS treatment was significantly reduced by MSL (Supplementary Fig. 7a). To study the mechanism of these findings, we studied NF-κB activation that could be reduced by calcineurin through TLR4, MyD88, or TRIF binding[32,33]. We observed that IκBα phosphorylation, disappearance of IκBα, p65 phosphorylation and NF-κB reporter activity induced by LPS were notably attenuated by MSL (Supplementary Fig. 7b, c), which indicates reduced NF-κB activation by MSL and suggests that MSL attenuates cytokine release through both autophagy-dependent and autophagy-independent mechanisms.

**Improved metabolic profile of obese mice by MSL**. We next studied whether autophagy enhancement by MSL could improve the metabolic profile of obese mice in vivo. Administration of 50 mg/kg MSL to *ob/ob* mice, a genetic mouse model of obesity with a *leptin* mutation[34], 3 times a week for 8 weeks significantly reduced nonfasting and fasting blood levels (Fig. 4a, b) without changes in food intake or body weight (Supplementary Fig. 8a, b). Intraperitoneal glucose tolerance test (IPGTT) and insulin tolerance test (ITT) showed significantly improved glucose tolerance and insulin sensitivity, respectively (Fig. 4c, d), with reduced area under the curves (AUCs) (Supplementary Fig. 8c, d). The glucose profile of lean mice was not affected by MSL (Fig. 4a–d and Supplementary Fig. 8c, d). Improved glucose profile after 8 weeks of MSL administration was accompanied by increased autophagic flux in the liver of *ob/ob* mice determined after leupeptin clamping (Supplementary Fig. 8e). Expression of TFEB-regulated genes such as *Uvrag*, *Clcn7*, *Atp6v0e1*, and *Tfeb* itself was significantly increased in the liver of MSL-treated mice (Supplementary Fig. 8f), showing enhanced expression of lysosomal genes and autophagy genes[35] after treatment with MSL for 8 weeks.

In the liver, fatty change and triglyceride (TG) accumulation were reduced by administration of MSL to *ob/ob* mice for 8 weeks (Fig. 4e, f). Serum aspartate transaminase (ASL) and alanine transaminase (ALT) levels were also decreased by MSL treatment (Fig. 4g), indicating decreased liver damage. In adipose tissue, the number of crown-like structures (CLSs), a marker of metabolic inflammation, was significantly decreased after 8 weeks of MSL administration (Fig. 4h, i). Real-time RT-PCR demonstrated significantly reduced expression of inflammatory genes such as *Tnfα*, *Il-6*, *pro-Il-1β*, and *F4/80* (Fig. 4j), which might be partly attributable to autophagy-independent mechanisms such as reduced NF-κB activation. Western blot analysis demonstrated that MSL treatment attenuated inflammasome activation in adipose tissue of *ob/ob* mice, as shown by reduced caspase 1 activation and IL-1β maturation (Fig. 4k). We next employed a diet-induced obesity model which is more physiological compared to *ob/ob* mouse model. MSL administration for 8 weeks appeared to reduce nonfasting blood glucose level and glucose intolerance in mice fed high-fat diet (HFD); however, statistical significance was not achieved for most comparisons except a certain point during IPGTT (Supplementary Fig. 9a–f).

**Metabolic effects of chemically modified autophagy enhancer**. One of the reasons the metabolic profile of HFD-fed mice was not significantly improved by MSL administration in vivo could be the poor microsomal stability of MSL (less than 10% remaining after 30 min incubation with human liver microsome). Thus, we chemically modified MSL to make more efficacious and druggable compounds. Among several derivatives, we selected a chemical (MSL-7) with improved microsomal stability (90.5% remaining after 30 min) (Fig. 5a). We confirmed that MSL-7 induced formation of autophagolysosome, TFEB nuclear translocation, and calcineurin activation in a dose-dependent manner (Fig. 5b, c and Supplementary Fig. 3a, b). MSL-7 also increased autophagic activity, reduced TFEB phosphorylation at S142 and bound to calcineurin A, protecting its degradation by pronase in DARTS assay, similar to MSL (Supplementary Fig. 3c and Supplementary Fig. 10a, b). In contrast, MSL derivatives (#9-3 and #9-4) that did not induce autophagy (data not shown) did not protect calcineurin A from degradation by pronase in DARTS assay (Supplementary Fig. 10b), suggesting the importance of calcineurin binding in autophagy induction by MSL or its derivatives. When we studied mTOR inhibition to confirm the absence of mTOR inhibition by MSL-7 employing Western blot analysis, phosphorylation of mTOR or S6K1 was not inhibited by MSL-7 or MSL, while mTOR or S6K1 activation was markedly inhibited by rapamycin or Torin-1 (Supplementary Fig. 10c). Phosphorylation of 4EBP1, another target of mTOR, was also not inhibited by MSL-7 or MSL, while 4EBP1 phosphorylation was markedly inhibited by Torin-1 but not by rapamycin, consistent with a previous paper[36] (Supplementary Fig. 10c).

Similar to MSL, MSL-7 expedited in vitro clearance of lipid, which was inhibited by lalistat 2, bafilomycin A1, or *Atg7* KO (Supplementary Fig. 5b, c). MSL-7 also attenuated IL-1β release or inflammasome activation by treatment with PA + LPS in vitro (Supplementary Fig. 6a, b), again similar to MSL. Mitochondrial ROS accumulation, reduced mitochondrial potential, and suppressed ATP-coupled mitochondrial oxygen consumption by PA + LPS combination were also reversed by MSL-7 (Supplementary Fig. 6c–e). Furthermore, mRNA levels of pro-IL-1β, TNFα, and IL-6 after treatment with LPS were significantly reduced by MSL-7 (Supplementary Fig. 7a). IκBα phosphorylation, disappearance of IκBα, p65 phosphorylation, and NF-κB reporter activity induced by LPS were also attenuated by MSL-7 (Supplementary Fig. 7b, c), which suggests that MSL-7 attenuates cytokine release through both autophagy-dependent and autophagy-independent mechanisms, similar to MSL.

We next studied the in vivo effect of MSL-7. When administered to *ob/ob* mice, MSL-7 significantly reduced nonfasting blood glucose level without changes in body weight (Fig. 5d, e). IPGTT and ITT demonstrated significantly improved glucose tolerance and insulin sensitivity, respectively, with reduced AUCs (Fig. 5f–i). When diet-induced obesity model was employed instead of *ob/ob* mice, administration of MSL-7 for 8 weeks significantly reduced the nonfasting blood glucose level of HFD-fed mice without changes in body weight (Fig. 6a, b). IPGTT and ITT also showed significantly improved glucose tolerance and insulin sensitivity, respectively, with reduced AUCs after MSL-7 treatment of HFD-fed mice (Fig. 6c–f), indicating that chemically modified autophagy enhancer can improve the

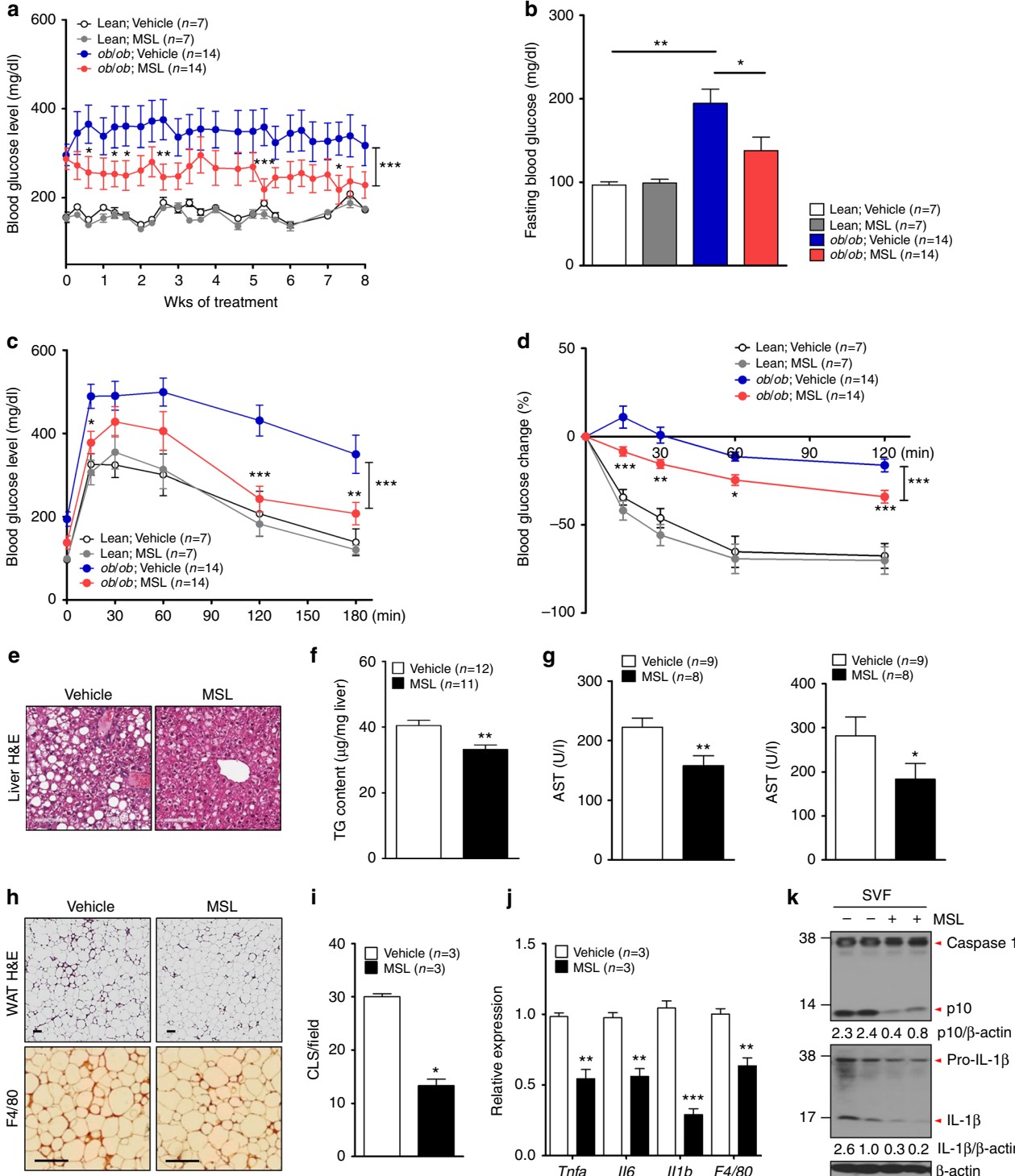

**Fig. 4** Improved metabolic profile of *ob/ob* mice by administration of autophagy enhancer. **a** Nonfasting blood glucose level of *ob/ob* mice treated with MSL ($F = 178.3$, df = 1). **b** Fasting blood glucose level after MSL treatment for 8 weeks ($F = 8.3$, df treatment = 3, df residual = 38). **c** IPGTT after the same treatment ($F = 49.5$, df = 1). **d** ITT after the same treatment ($F = 48.4$, df = 1). **e** Hepatic H&E-stained sections from MSL-treated *ob/ob* mice. **f** Hepatic triglyceride content ($t = 3.5$, df = 22). **g** Serum AST/ALT levels ($t = 3.7$, df = 15 for AST; $t = 2.3$, df = 15 for ALT). **h** H&E staining (upper) and immunohistochemistry using F4/80 antibody (lower) of adipose tissue from *ob/ob* mice treated with MSL for 8 weeks. **i** Numbers of CLSs in F4/80 antibody-stained sections ($t = 9.4$, df = 2). **j** Expression of cytokines in adipose tissue determined by real-time RT-PCR ($t = 6.3$, df = 4 for *Tnfa*; $t = 6.5$, df = 4 for *Il-6*; $t = 11.4$, df = 4 for *Il-1β*; $t = 5.5$, df = 4 for *F4/80*). **k** Inflammasome activation in adipose tissue examined by Western blot analysis. Numbers below immunoblot bands indicate fold changes of cleaved caspase 1 (p10) or mature IL-1β bands normalized to β-actin bands. All data in this figure are the means ± s.e.m. from ≥3 independent experiments performed in triplicate (scale bar, 100 μm). *$P < 0.05$, **$P < 0.01$ and ***$P < 0.001$ by two-way ANOVA with Bonferroni's post-hoc test (**a**, **c**, **d**), one-way ANOVA with Tukey's post-hoc test (**b**) or two-tailed Student's *t*-test (**f**, **g**, **i**, **j**)

glucose profile not only of *ob/ob* mice but also of mice with diet-induced obesity. MSL-7 did not affect the glucose profile of lean or chow-fed mice (Fig. 5d–i and Fig. 6a–f). Levels of fasting blood glucose, serum insulin, C-peptide or leptin, and HOMA-IR index representing insulin resistance that were increased in HFD-fed mice were significantly lowered by MSL-7 administration for 8 weeks (Fig. 6g–k). Serum level of nonesterified fatty acids

(NEFA) that was increased in HFD-fed mice was also lowered by MSL-7 administration for 8 weeks, while the difference was not statistically significant (Fig. 6l). In contrast, reduced serum adiponectin level in HFD-fed mice was increased by MSL-7 administration for 8 weeks (Fig. 6m).

We next studied whether metabolic improvement by in vivo administration of MSL-7 is associated with TFEB activation. In

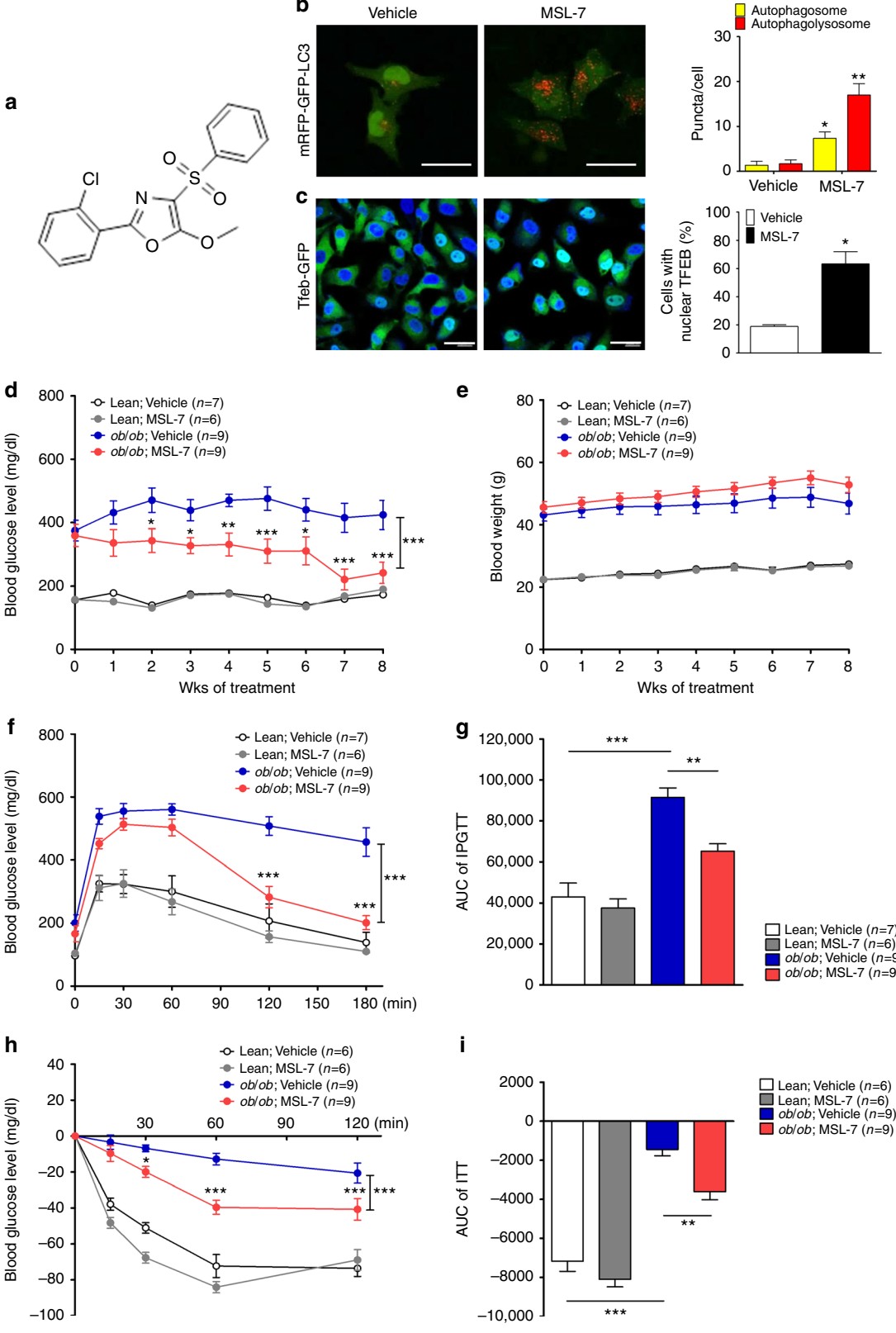

the liver of mice fed HFD, TFEB phosphorylation at S142 was increased, suggesting impaired TFEB signaling in vivo. Administration of MSL-7 for 8 weeks significantly reduced TFEB phosphorylation at S142 in the liver of HFD-fed mice (Supplementary Fig. 11), suggesting enhanced TFEB signaling by MSL-7 in vivo. We also investigated whether knockdown of *Tfeb* eliminates the beneficial effect of MSL-7 in vivo to obtain genetic evidence that MSL-7 improve the metabolic profile of obese mice by activating the TFEB pathway. In vivo administration of 1 mg/kg *Tfeb* siRNA through the tail vein every 7–10 days significantly reversed MSL-7-induced improvement of glucose profile in HFD-fed mice without significant change of body weight (Supplementary Fig. 12a, b), indicating that the in vivo metabolic effect of MSL-7 is through activation of TFEB. Significant *Tfeb* knockdown by *Tfeb* siRNA was confirmed in vitro (Supplementary Fig. 12c) and also in vivo (Supplementary Fig. 12d), while the in vivo effect was relatively weaker compared to the in vitro effect because liver tissues were obtained at the end of the in vivo experiment (i.e., 11 days after the last *Tfeb* siRNA injection when the in vivo effect waned). IPGTT and ITT also showed that in vivo *Tfeb* knockdown partially reversed improved glucose tolerance and insulin sensitivity by MSL-7 administration (Supplementary Fig. 12e–h). The expression of gluconeogenesis genes such as *glucose-6-phosphatase, phosphoenolpyruvate carboxykinase* (PEPCK), *fructose 1,6 bisphosphatase,* or *pyruvate carboxylase*, which was increased in the liver of HFD-fed mice, was reduced by MSL-7 (Supplementary Fig. 13), probably due to improved insulin signaling.

When we studied the expression of TFEB downstream genes after in vivo administration of MSL-7, the expression of *Lc3, Beclin-1, Lamp1, CtsA,* and *CtsD* was increased or derepressed in the liver after 8 weeks of MSL-7 administration (Supplementary Fig. 14a). The expression of *Tfeb* itself was increased after MSL-7 administration (Supplementary Fig. 14a), consistent with a previous paper reporting that TFEB induces the expression of *Tfeb* itself[35]. In skeletal muscle, the expression of most *Tfeb* target genes was not increased after 8 weeks of MSL-7 administration to HFD-fed mice (Supplementary Fig. 14b). On the other hand, the expression of mitochondrial genes such as *Mfn1, Mfn2, Nrf-1, Tfam, CoxI, CoxII, CoxIV, CoxVa, Cox8b,* or *Dlat* was upregulated by MSL-7 (Supplementary Fig. 14c), which shows differential regulation of TFEB target genes depending on tissue, in line with a previous paper[37]. In epididymal white adipose tissue (eWAT), the expression of *Tfeb* and several *Tfeb* downstream genes was enhanced by HFD feeding alone and not further increased by administration of MSL-7 (Supplementary Fig. 14d), which may be due to upregulation of lysosomal genes in adipocytes or adipose tissue macrophages as an adaptation to a local lipid-rich environment[38]. However, the number of red puncta in the eWAT of HFD-fed *GFP-RFP-LC3* mice representing the autophagolysosome was significantly increased after MSL-

7 administration (Supplementary Fig. 15), indicating that MSL-7 indeed exerted its effects on adipose tissue. Dissociation of *Tfeb* expression and autophagolysosome number could be due to the defective fusion between autophagosome and lysosome in HFD-fed mice[39] leading to no increase of autophagosome number despite increased *Tfeb* expression, which might be resolved after MSL-7 administration. The expression of autophagy genes and lysosomal genes in the pancreas was not significantly changed by MSL-7 treatment (Supplementary Fig. 14e).

When the changes in weights of metabolic organs were examined, weight of the liver was reduced to a small but significant degree after in vivo administration of MSL-7 for 8 weeks probably due to reduced lipid content (Supplementary Fig. 16a), consistent with previous data showing reduced liver weight after genetic overexpression of *Tfeb* in the liver[35]. In contrast, epididymal fat weight was increased after in vivo administration of MSL-7 (Supplementary Fig. 16a), which may be due to increased adipogenesis by *Tfeb* activation in adipose tissue[40]. The expression of PCG-1α, a master regulator of mitochondrial biogenesis, appeared to be increased in brown adipose tissue and subcutaneous white adipose tissue (sWAT) after MSL-7 administration; however, the expression of thermogenesis-related genes such as *Ucp1, deiodinase2,* or *Elovl3* was not significantly or only marginally increased by MSL-7 administration (Supplementary Fig. 16b, c). The expression of lipogenesis-related genes such as *Pparγ2, Scd1, C/ebpα, C/ebpβ* or *Fasn* was significantly increased after in vivo treatment with MSL-7 in eWAT but not in sWAT (Supplementary Fig. 16d, e), which might be related to the increased epididymal fat weight after MSL-7 treatment. The expression of *Angptl4*, a lipoprotein lipase inhibitor, was increased probably as a reactive change to increased lipogenic gene expression. The expression of *Glut4* or *Irs1*, which are critical in insulin sensitivity, was not significantly changed by MSL-7 administration (Supplementary Fig. 16d, e). The size of adipocytes was not significantly changed after in vivo administration of MSL-7 (Supplementary Fig. 16f).

We finally studied the possible toxicity of autophagy-enhancer small molecules. Administration of MSL or MSL-7 for 8 weeks did not affect hemogram or blood chemistry of *ob/ob* mice except improved metabolic profile and decreased liver enzyme levels (Supplementary Table 1). Biopsy of the major organs revealed no abnormalities except improved fatty liver changes (Supplementary Fig. 17), supporting no significant toxicity and potential druggability of MSL or MSL-7.

## Discussion

We here developed new potential therapeutics that can enhance autophagic activity and improve the metabolic profile of mice with metabolic syndrome and obesity. These results are consistent with previous reports that systemic autophagy deficiency aggravated metabolic syndrome[8], and that genetic *Atg5* overexpression

**Fig. 5** Improved metabolic profile after in vivo administration of autophagy enhancer with increased microsomal stability (MSL-7) to *ob/ob* mice. **a** Chemical structure of chemically modified autophagy enhancer, MSL-7, with increased microsomal stability [2-(2-chlorophenyl)-5-methoxy-4-(phenylsulfonyl)oxazole]. **b** HeLa cells transfected with tandem *mRFP-GFP-LC3* construct were treated with MSL-7 for 1 h, and were subjected to confocal microscopy (left panel). The numbers of yellow and red punctae representing autophagosomes and autophagolysosomes, respectively, were counted (right) ($t = 3.5$, df = 4 for autophagosome; $t = 5.8$, df = 4 for autophagolysosome). **c** *TFEB-GFP*-transfectant HeLa cells were treated with MSL-7 for 4 h, and were subjected to confocal microscopy (left panel). The number of cells with nuclear TFEB was counted (right) ($t = 5.0$, df = 2). **d, e** Eight-week-old male *ob/ob* mice or 8-week-old male C57BL/6 mice were treated with 50 mg/kg MSL-7 3 times a week for 8 weeks, and nonfasting blood glucose level (**d**) and body weight (**e**) were monitored ($F = 40.9$, df = 1 for **d**). **f** IPGTT was conducted after in vivo administration of MSL-7 for 8 weeks ($F = 62.3$, df = 1). **g** AUC of the IPGTT curve in **f** ($F = 25.3$, df treatment = 3, df residual = 27). **h** ITT was conducted after in vivo administration of MSL-7 for 8 weeks ($F = 50.1$, df = 1). **i** AUC of the ITT curve in **h** ($F = 57.7$, df treatment = 3, df residual = 26). All data in this figure are the means ± s.e.m. from ≥3 independent experiments performed in triplicate (scale bar, 20 μm). *$P < 0.05$, **$P < 0.01$ and ***$P < 0.001$ by one-way ANOVA with Tukey's post-hoc test (**g, i**), two-way ANOVA with Bonferroni's post-hoc test (**d, f, h**) or two-tailed Student's *t*-test (**b, c**)

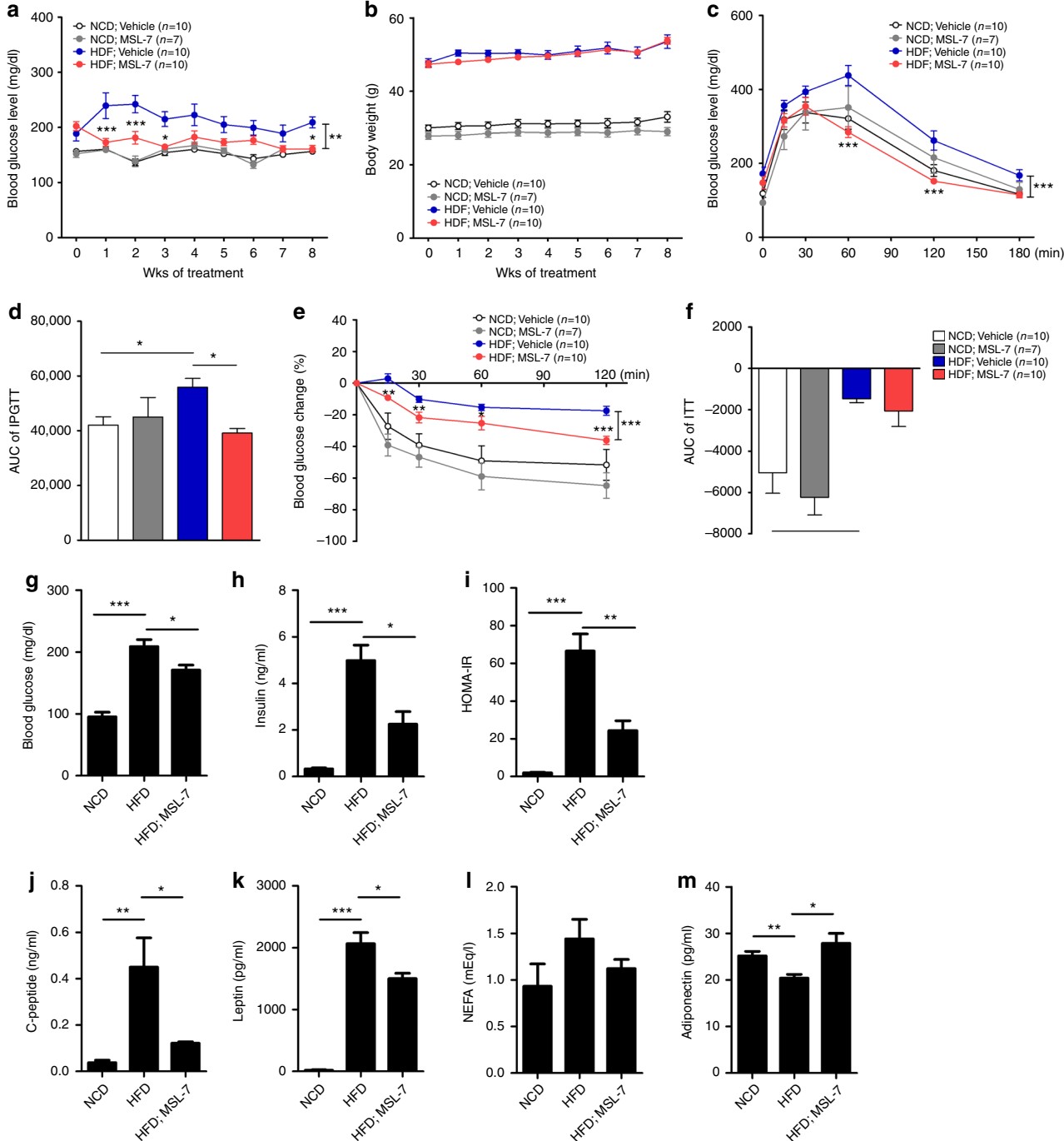

**Fig. 6** Improved metabolic profile of HFD-fed mice after administration of chemically modified autophagy enhancer, MSL-7. **a**, **b** Eight-week-old male C57BL/6 mice were fed HFD or normal chow diet (NCD) for 8 weeks, and then treated with 50 mg/kg MSL-7 3 times a week for 8 weeks. Nonfasting blood glucose level (**a**) and body weight (**b**) were monitored ($F = 56.5$, df = 1 for **a**). **c** IPGTT was conducted after in vivo administration of MSL-7 for 8 weeks ($F = 52.7$, df = 1). **d** AUC of the IPGTT curve in **c** ($F = 4.2$, df treatment = 3, df residual = 33). **e** ITT was conducted after in vivo administration of MSL-7 for 8 weeks ($F = 52.6$, df = 1). **f** AUC of the ITT curve in **e** ($F = 10.3$, df treatment = 3, df residual = 33). **g**–**m** Metabolic parameters of mice treated with MSL-7. Fasting blood glucose level (**g**) and serum levels of insulin (**h**), C-peptide (**j**), leptin (**k**), NEFA (**l**), and adiponectin (**m**) were determined. HOMA-IR was calculated (**i**). All data in this figure are the means ± s.e.m. from ≥ 3 independent experiments performed in triplicate. *$P < 0.05$, **$P < 0.01$ and ***$P < 0.001$ by two-way ANOVA with Bonferroni's post-hoc test (**a**, **c**, **e**), one-way ANOVA with Tukey's post-hoc test (**d**, **f**) or two-tailed Student's t-test (**g**–**m**)

improved the metabolic profile of aged mice[9]. Our data are also in line with a report that cardiac autophagy plays a protective role in the ischemic cardiac disease of mice with diet-induced obesity[41] or that spermidine improved cardiac function and extended the lifespan of aged mice by enhancing autophagy[42]. Accordingly, a

couple of autophagy enhancers have been employed against metabolic syndrome in previous studies[8,43]. However, they might not be good candidates for therapeutic agents, as the autophagy-enhancing activity of trehalose has been questioned in a recent paper[44] and imatinib may have side effects precluding its use in

metabolic syndrome[45]. Thus, our small-molecule autophagy enhancers could be novel candidates for therapeutic agents against metabolic syndrome without adverse effects.

Metabolic improvement by MSL and MSL-7 through calcineurin activation and TFEB dephosphorylation is similar to the effects of adenoviral expression of *Tfeb*[35] or its homolog *Tfe3*[46] in vivo. Causality between TFEB activation and metabolic improvement was confirmed by reversal of metabolic changes after MSL-7 administration by in vivo knockdown of *Tfeb*. MSL-7 administration also reversed TFEB S142 phosphorylation in the liver of HFD-fed mice. Maximum plasma concentrations of MSL and MSL-7 (Cmax) after 50 mg/kg administration were 0.21 μg/ml (0.61 μM) and 0.57 μg/ml (1.63 μM), respectively (Song JS et al., unpublished data). At such concentrations, proportions of cells with TFEB nuclear translocation would be small (see Supplementary Fig. 3). We hypothesize that such a small degree of nuclear translocation and S142 dephosphorylation of TFEB may explain the metabolic effect of MSL or MSL-7 in vivo. While uncontrolled or excessive TFEB activation for a prolonged period can be oncogenic[47], TFEB activation to an appropriate degree and duration may be beneficial to host metabolism and nutrient homeostasis since lysosomal function[48] and autophagic or mitophagic activity[49,50] decline markedly with aging. Calcineurin activation has been reported to directly enhance β-cell function or muscle endurance capacity[51,52]. Additionally, calcineurin activation may improve the metabolic profile by upregulating autophagic activity and preserving organelle function.

It is well established that two key components of type 2 diabetes are insulin resistance and β-cell failure. However, molecular and cellular pathogeneses of type 2 diabetes leading to insulin resistance and β-cell failure are not clearly delineated[28,53–55]. Obesity and lipid overload are major risk factors predisposing to the development of diabetes. At the organelle level, dysfunction or stress of endoplasmic reticulum (ER) and mitochondria is an important etiological component in the development of diabetes[56,57]. Autophagy deficiency due to aging, lipid, or other causes[49,50,58] can lead to dysfunction of ER or mitochondria[2]. Furthermore, autophagy deficiency compromises adaptation to metabolic stress or clearance of accumulated lipid[6,8,29]. Thus, autophagy deficiency in obesity or aging could, in part, be an underlying cause of ER or mitochondrial dysfunction associated with diabetes and metabolic syndrome. Enhancement of autophagic activity might be a novel therapeutic approach against metabolic disorders by targeting fundamental pathogenesis of these diseases, which could be different from conventional modalities of diabetes treatment that address abnormal molecular or cellular processes emanating from the underlying defects such as decreased autophagic activity and ensuing organelle dysfunction.

## Methods

**Screening of autophagy enhancer.** HepG2 cells in a 100-mm culture dish were stably transfected with 5 μg of pRLuc(C124A)-LC3(WT) or a mutant harboring the G120A substitution [pRLuc(C124A)-LC3(G120A)] that is resistant to proteolytic cleavage and inhibits conversion of LC3-I to LC3-II, using 10 μl lipofectamine[14]. Stable transfectants were selected by culturing in the presence of 400 μg/ml G418 (Invitrogen), and clones that showed normalized wild/mutant luciferase ratio <0.6 after treatment with 250 nM rapamycin for 6 h were isolated for library screening.

**Luciferase assay.** After treatment of stable HepG2 transfectants expressing pRLuc (C124A)-LC3(WT) or pRLuc(C124A)-LC3(G120A) with 50 μM chemicals for 24 h, cells were lysed in a Passive Lysis Buffer (Promega). After addition of an assay buffer containing 10 μM coelenterazine (Promega), *Renilla* luciferase activity was measured using a microplate luminometer (Berthold Technologies).

**Cell culture and drugs treatment.** HeLa (Korean Cell Line Bank), stable *Tfeb*-GFP-transfected HeLa, CRISPR/Cas9 *Tfeb* knockout HeLa cells, Δ*CnA*-transfected HeLa cells and SK-Hep1 cells (Korean Cell Line Bank) were cultured in DMEM

supplemented with 10% FBS. HeLa cells were employed for study of lipid metabolism because HeLa cells express ATGL and HSL. Hepa1c1c7 (Korean Cell Line Bank) and HepG2 cells (Korean Cell Line Bank), which are well-characterized hepatocyte cell lines and have long been employed for metabolic studies, were grown according to the protocols provided by ATCC. All cells were free of mycoplasma contamination. All cells were routinely monitored for morphologic or growth changes to prevent cross contamination or genetic drift. For drug treatment, the following concentrations were used: MSL, 100 μM unless stated otherwise; MSL-7, 100 μM unless stated otherwise; rapamycin, 250 nM; bafilomycin, 100 nM unless stated otherwise; cyclosporin A, 10 μM; and FK506, 5 μM. To study the effect of autophagy enhancer on translocation and phosphorylation of TFEB or calcineurin activity, cells were treated with MSL or MSL-7 for 4 h. For lipid loading, cells were treated with a combination of PA (400 μM) + OA (800 μM) for 24 h. PA solution was made as previously reported[27]. Briefly, PA stock solution (50 mM) was prepared by dissolving in 70% ethanol and heating at 50 °C. OA stock solution (500 mM) was prepared by dissolving in 100% ethanol. The working solution was made by diluting PA stock and/or OA stock solution in 2% fatty acid-free BSA-DMEM. As a control solution, 2% fatty acid-free BSA-DMEM was employed. To study the clearance of lipid by autophagy enhancer, lipid-loaded cells were treated with MSL for 16 h. To study colocalization of lipid with autophagosome or lysosome marker, lipid-loaded cells treated with MSL for 1 h were employed to observe LD before disappearance. Pretreatment with calcineurin inhibitors was performed since 1 h before test treatment. MSL (ChemBridge) and MSL-7 were dissolved in DMSO to make 10 mM and directly diluted to the final concentrations in culture medium for in vitro experiments. For in vivo administration, MSL and MSL-7 were dissolved in DMSO to yield a 50 mg/ml stock solution, which was diluted with PBS to 5 mg/ml before injection. All in vitro experiments were repeated at least three times.

**Cell death assay.** When treatment with chemicals was completed, medium was removed and 0.5 mg/ml 3-[4,5-dimethylthiazol-2-yl]-2,5-diphenyltetrazolium bromide (MTT) added. After incubation at 37 °C for 2 h in a CO$_2$ incubator and a brief centrifugation, supernatant was carefully removed. DMSO was then added to dissolve insoluble crystals completely, followed by measuring absorbance at 540 nm using Thermomax microplate reader (Molecular Devices)[59]. Lipotoxicity was determined by measuring lactate dehydrogenase release from damaged cells using a kit (Promega).

**Transfection and plasmids.** Cells were transiently transfected with plasmids such as 3x*Flag-Tfeb*, -*Tfeb*(S142D), tandem *mRFP-GFP-LC3*, *mRFP-LC3*, *GCaMP3-ML1*, *HA-ΔCnA*, or -*ΔCnA*-H151Q using lipofectamine 2000 (Invitrogen) according to the manufacturer's protocol. When *HA-ΔCnA* was employed, the same dose of regulatory calcineurin B subunit (*CnB*) was transfected together[24].

**Imaging and image quantification.** Imaging was conducted using an LSM780 confocal microscope (Zeiss). Counting of the numbers of acidic vesicle, autophagic puncta, and LD was performed using ImageJ. To visualize LD, HeLa cells were stained with 20 μg/ml BODIPY493/503 (Invitrogen) for 20 min. To observe acidic vesicular organelles, HeLa cells were treated with autophagy enhancer for 24 h and then stained with 5 μg/ml AO (Invitrogen) for 10 min.

**Antibodies and Western blot analysis.** Cells or tissues were solubilized in a lysis buffer containing protease inhibitors. Protein concentration was determined using the Bradford method. Samples (10–30 μg) were separated on 4–12% Bis–Tris gel (NUPAGE, Invitrogen) or 8–15% SDS-PAGE gel, and transferred to PVDF or nitrocellulose membranes for Western blot analysis using the ECL method (Pierce). Antibodies against the following proteins were used: LC3 (Novus NB100–2331, 1:1,000), p62 (Progen GP62-C, 1:1,000), β-actin (Santa Cruz sc47778, 1:5,000), FLAG (Sigma-Aldrich F3165, 1:2,000), HA (Cell Signaling #2367, 1:1,000), S6K1 (Cell Signaling #9202, 1:1,000), phospho-S6K1 (Cell Signaling #9206, 1:1,000), mTOR (Cell Signaling #2983, 1:1,000), phospho-mTOR (Cell Signaling #2971, 1:1,000), phospho-4EBP1 (Cell Signaling #2855, 1:1,000), 4EBP1 (Cell Signaling #9644, 1:1,000), TFEB (Bethyl Laboratories A303-673A, 1:1,000), phospho-S142-TFEB (Millipore ABE1971, 1:1,000), LAMP1 (Abcam ab24170, 1:250), caspase 1 (Santa Cruz sc514, 1:1,000), IL-1β (Santa Cruz sc7884, 1:1,000), phospho-IκBα (Santa Cruz sc9246, 1:1,000), IκBα (Santa Cruz sc307, 1:1,000), phospho-p65 (Cell Signaling #3033, 1:1,000), or p65 (Cell Signaling #8242, 1:1,000). Densitometry of the protein bands was performed using ImageJ. Images of original Western blotting are shown in Supplementary Fig. 18.

**RNA extraction and real-time RT-PCR.** Total RNA was extracted from cells or tissues using TRIzol (Invitrogen), and cDNA was synthesized using MMLV Reverse Transcriptase (Promega) according to the manufacturer's protocol. Real-time RT-PCR was performed using SYBR green (Takara) in ABI PRISM 7000 (Applied Biosystems). All expression values were normalized to *S18* mRNA level. Primer sequences are listed in Supplementary Table 2.

**GCaMP3 Ca$^{2+}$ imaging**. HeLa cells were grown on 15 mm coverslips and transfected with a plasmid encoding a perilysosomal GCaMP3-ML1 Ca$^{2+}$ probe. After 48 h, lysosomal Ca$^{2+}$ release was measured in a basal Ca$^{2+}$ solution containing 145 mM NaCl, 5 mM KCl, 3 mM MgCl$_2$, 10 mM glucose, 1 mM EGTA and 20 mM HEPES (pH 7.4) with or without MSL, by monitoring fluorescence intensity at 470 nm with a LSM780 confocal microscope (Zeiss). GPN (200 μM) was used as a positive control for induction of Ca$^{2+}$ release from lysosome. Ionomycin (1 μM), a calcium ionophore, was employed to induce a maximal Ca$^{2+}$ response.

**Calcineurin phosphatase activity**. The phosphatase activity of calcineurin was determined using a calcineurin phosphatase activity assay kit (Abcam), according to the manufacturer's protocol.

**ELISA**. Primary peritoneal macrophages were isolated from C57BL/6 mice using 3.85% thioglycollate solution, and treated with 400 μM PA and/or 100 ng/ml LPS. After incubation for 24 h, IL-1β content in culture supernatants was determined using a mouse ELISA kit (R&D Systems).

**Mitochondrial changes**. To determine mitochondrial potential, peritoneal macrophages were stained with 1 μM each of MitoTracker Green and MitoTracker Red (Invitrogen) at 37 °C for 25 min. Cells were suspended in PBS-1% FBS for analysis on a FACSVerse (BD Biosciences) using the FlowJo software (TreeStar). To measure mitochondrial ROS content, cells were incubated with 5 μM MitoSox (Invitrogen) at 37 °C for 5 min, and FACS analysis was performed. Mitochondrial oxygen consumption rate (OCR) was measured using XF24 analyzer (Seahorse Bioscience). Peritoneal macrophages were seeded onto XF24 well plates at a density of $1 \times 10^6$ cells/well and treated with 100 ng/ml LPS and/or 200 μM PA as indicated in the text. After 24 h of incubation, cells were washed and placed in XF medium. OCR was measured under basal conditions and after addition of 1 μM oligomycin to calculate ATP-coupled oxygen consumption.

**Animals**. Eight-week-old male *ob/ob* mice (Jackson Laboratory) were maintained in a 12-h light/12-h dark cycle and fed a chow diet. Fifty mg/kg MSL, MSL-7, or vehicle was administered intraperitoneally to 8-week-old male *ob/ob* mice 3 times a week for 8 weeks. During the observation period, mice were monitored for glucose profile and weighed. In experiments using diet-induced obesity models, 8-week-old male C57BL/6 mice were fed HFD for 8 weeks, and then treated with 50 mg/kg MSL or MSL-7 3 times a week for 8 weeks together with HFD feeding. Plasma concentrations of MSL or MSL-7 after in vivo administration were determined by LC-MS/MS. For in vivo clamping of the lysosomal steps of autophagy, 30 mg/kg leupeptin (Sigma) was administered through tail vein. *CAG-RFP-EGFP-LC3*-transgenic mice (Jackson Laboratory) were fed HFD for 8 weeks, and then treated with 50 mg/kg MSL-7 3 times a week for 8 weeks together with HFD feeding. Tissue sections prepared as described[60] were subjected to fluorescent microscopy after DAPI staining to identify RFP and GFP puncta. Sample size of mouse experiments was chosen based on previous in vivo data using *ob/ob* mice or HFD-fed mice with α error of 0.05, β error of 0.2, and effect size of 0.25. Mice showing nonfasting blood glucose level above 500 mg/dl before in vivo experiment or apparently abnormal sick mice were excluded from the study. Mice experiments were not randomized and conducted without employing the blinding technique

All animal experiments were conducted in accordance with the Public Health Service Policy in Humane Care and Use of Laboratory Animals. Mouse experiments were approved by the IACUC of the Department of Laboratory Animal Resources of Yonsei University College of Medicine, an AAALAC-accredited unit. The number of mice used is shown in each figure.

**Glucose tolerance test and insulin tolerance test**. IPGTT was performed by intraperitoneal injection of 1 g/kg glucose after overnight fasting. Blood glucose concentrations were determined using an One Touch glucometer (Lifescan) before (0 min) and 15, 30, 60, 120, and 180 min after glucose injection. ITT was conducted by injecting 0.75 U/kg of regular insulin intraperitoneally to fasted mice and measuring blood glucose levels at 0, 15, 30, 60, and 120 min. HOMA-IR was calculated using the following formula: (fasting insulin x fasting glucose)/22.5.

**Chemical synthesis of MSL**. A solution of methyl bromoacetate (10 g, 65.38 mmol) and benzenesulfinic acid, sodium salt (12.9 g, 78.4 mmol) in ethanol (200 ml) was refluxed overnight. Excess solvent was removed under reduced pressure. The mixture was dissolved in dichloromethane (400 ml) and washed with water (2 × 200 ml) and brine (150 ml). The organic layer was dried over anhydrous Na$_2$SO$_4$ and concentrated under reduced pressure to give title compound (13.5 g, 96%). The compound was used in the next step without further purification.

To a stirred solution of methyl 2-(phenylsulfonyl) acetate (13.5 g, 67.7 mmol) and 4-acetamidobenzenesulfonyl azide (16.65 g, 69.31 mmol) in acetonitrile (500 ml) at 0 °C, was added triethylamine (7.0 g, 69.3 mmol) dropwise. The reaction mixture was stirred at room temperature for 24 h. The reaction mixture was filtered and washed solid with ethyl acetate thoroughly. The filtrate was concentrated *in vacuo*. The resulting residue was purified by column chromatography using ethyl acetate and *n*-hexane to give the title compound as a pale yellow solid (15 g, 99%).

To a refluxing solution of 2-chlorobenzonitrile (600 mg, 4.36 mmol) and rhodium(I1) acetate (38.55 mg, 0.087 mmol) in chloroform (10 ml) was added a solution of methyl 2-diazo-2-(phenylsulfonyl) acetate (1.15 g, 4.8 mmol) in chloroform (10 ml). After the addition was finished, the reaction mixture was maintained under reflux condition for 3 h. Reaction mixture was cooled and concentrated under reduced pressure. The residue was purified by column chromatography to afford title compound as a white solid (1.4 g, 92%). $^1$H NMR (300 MHz, DMSO-d$_6$): δ 7.99–7.87 (m, 3H), 7.76–7.59 (m, 4H), 7.58–7.44 (m, 2H), 4.25 (s, 3H).

**In vivo transfection**. *Tfeb* siRNA (CCAACCUGUCCAAGAAGGA)[61] (Sigma) was administered through tail vein together with Invivofectamine® 3.0 (ThermoFisher) every 7–10 days for a total of 3 times since in vivo effect of Invivofectamine® 3.0 wanes after 10 days according to the manufacturer's instruction. Efficacy of *Tfeb* siRNA knockdown was evaluated by real-time RT-PCR using mRNA from cells or liver tissues and specific primers.

**NF-κB reporter assay**. NF-κB reporter activity was determined using pELAM-luciferase NF-κB reporter construct, as previously described[62]. In short, primary peritoneal macrophages were co-transfected with 0.2 μg of pELAM-luc and 0.02 μg of pRL-TK plasmids using Lipofectamine 2000. Twenty-four h later, transfected cells were treated with 100 ng/ml LPS with or without pretreatment with 100 μM MSL or MSL-7. After incubation for 6 h and cell lysis in Passive Lysis Buffer (Promega), reporter gene activities were determined using Dual-Luciferase assay system (Centro LB 960) and presented as firefly luciferase activity normalized to *Renilla* luciferase activity.

**Histology and immunohistochemistry**. Tissue samples were fixed in 10% buffered formalin and embedded in paraffin. Sections of 5 μm thickness were stained with H&E for morphometry, or immunostained with F4/80 antibody (Millipore) to detect macrophage aggregates surrounding adipocytes (CLSs). Adipocyte diameter was measured per each section using ImageJ.

**Blood chemistry and hemogram**. Blood chemistry was determined using a Fuji Dri-Chem analyzer. Hemogram was obtained from heparinized blood using a Hamevet950 Blood Analyzer (Drew Scientific).

**Stromal vascular fraction (SVF)**. To isolate SVF, epididymal adipose tissue was minced to ~2 mm pieces. After digestion in 2 mg/ml type 2 collagenase (Worthington) solution at 37 °C for 45 min, the tissue was centrifuged at 1000$g$ for 8 min. After filtration through a 70 μm mesh and lysis of red blood cells, SVF was suspended in PBS-2% BSA (Roche)-2 mM EDTA (Cellgro) for further experiments.

**Triglyceride measurement**. For biochemical measurement of hepatic TG content, the lipid was extracted from homogenized tissue using chloroform/methanol mixture (2:1). Lipid residue after evaporation was suspended in 1% Triton X-100 in 100% ethanol, and mixed with Free Glycerol Reagent containing lipase (Sigma). After incubation at 37 °C for 5 min, absorbance at 540 nm was measured for calculation of TG concentrations using a standard curve.

**DARTS assay**. HeLa cells were scraped and lysed with 8 M urea lysis buffer. After centrifugation for 15 min at 10,000$g$, supernatant was obtained and protein content was quantified using Bradford solution. Before drug treatment, protein concentration was diluted to 1 mg/ml. Samples were incubated with MSL of excessive concentration to maximize the interaction at 4 °C for 4 h. Samples were then treated with pronase (Roche) for 0, 5, 10, 15 min at 25 °C, and then subjected to Western blot analysis using anti-calcineurin A antibody.

**Liver microsomal stability**. The reaction mixture consisted of human liver microsomes (BD Gentest) in 100 mM potassium phosphate buffer (pH 7.4) and 10 μM test chemicals. After preincubation at 37 °C for 5 min, the reaction was initiated by adding NADPH regenerating solution (BD Biosciences). Samples (50 μl) were collected at 0 and 30 min. The reaction was terminated by adding 450 μl of ice-cold acetonitrile with imipramine (100 ng/ml, internal standard). After vortexing and centrifugation at 4 °C for 5 min at 13,000 rpm, the clear supernatant was collected, transferred to liquid chromatography (LC) vials, and analyzed by LC-MS/MS (Agilent 6460) for the quantification of the chemicals.

**Statistical analysis**. All values are expressed as the means ± s.e.m. of ≥3 independent experiments performed in triplicate. Two-tailed Student's *t*-test was used to compare values between two groups. One-way ANOVA with Tukey's test was used to compare values between multiple groups. Two-way repeated-measures ANOVA with Bonferroni's post-hoc test was employed to compare multiple repeated measurements between groups. Estimation of variation was evaluated by computing the sample standard deviation and error in each experiment. We did not observe notable differences in variances between groups or breach of normal

distribution precluding application of statistical analyses used in this study. *P* values <0.05 were considered to represent statistically significant differences.

**Data availability**. All data related to this manuscript and Supplementary Information are available from the corresponding author upon reasonable request.

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

## Acknowledgements

The authors thank M. Jäättelä for kind provision of p*RLuc*(C124A)-*LC3*(WT) and p*RLuc*(C124A)-*LC3*(G120A) constructs and J.S. Song for measurement of in vivo concentration of MSL or MSL-7. CRISPR/Cas9 *Tfeb*-knockout HeLa cells, *Tfeb-GFP*-transfectant HeLa cells, 3x*FLAG-Tfeb* WT/S142D mutant, Δcan/ΔCnA-H151Q, m*RFP-GFP*, and *GCaMP3-ML1* plasmid are gifts from R. Youle, E. Jho, A. Ballabio, L. Scorrano, T. Yoshimori, and H. Xu, respectively. This study was supported by Global Research Laboratory Grant (K21004000003-12A0500-00310) and Bio&Medical Technology Development Program (NRF-2015M3A9B6073846). M-S Lee and HJ Kwon are the recipient of UNIST Fund (2014M3A9D8034459 to M-S Lee) and NRF grants (M.-S. Lee: 2015K2A2A6002060; H.J. Kwon: 2015K1A1A2028365 and 2015M3A9C4076321).

## Author contributions

K.H.K. and M.-S.L. conceived the study. M.-S.L. designed the experiments. H.L., J.K, Y.-M.L., K.P, K.H.K., Y.E.J., H.-Y.H., D.J.L., H.P., H.J.K., and J.H.A. conducted the experiments. M.-S.L. wrote the manuscript with input from other authors.

## Additional information

**Competing interests:** The authors declare no competing interests.

