## [Peer Review File · Nature Communications]

Reviewers' comments:

Reviewer #1 (expert in autophagy, apoptosis, liver injury)(Remarks to the Author):

Metabolic syndrome and diabetes are important worldwide health issues currently. In this manuscript, authors identified a novel autophagy inducer from an autophagy screening and investigated its role and underlying mechanisms against metabolic syndrome in mice. They found this compound (MSL) activated TFEB likely through activation of phosphatase calcineurin but not mTOR and in turn enhanced hepatic autophagy, which might help to remove excess hepatic lipid droplets and also improve glucose tolerance and insulin sensitivity. While the data on MSL induced beneficial effects against metabolic syndrome in ob/ob mice and diet-induced obesity mice were promising, the mechanistic link and insights on TFEB and autophagy in vivo was relatively weak. Some of the data were also not convincing.

Major Concerns:

1. Despite that MSL improved metabolic syndrome in ob/ob mice. It was unclear whether there was a defect of TFEB signaling in the ob/ob mice or high fat diet-induced obesity. Although authors provided strong data to show MSL activated TFEB in cultured cells, not data to show MSL also activated TFEB in ob/ob mice or high fat diet-treated mice. In addition, whether knockdown of TFEB in ob/ob mice would eliminate MSL beneficial effects were not determined.
2. Multiple tissues including muscle, adipose and pancreatic tissues in addition to liver are all important for metabolic syndrome. However, it seemed that only the histological changes on these tissues were determined (Figure 4 e and h and supplemental figure 6) but the more important lysosomal biogenesis and autophagy activities were not determined in these tissues after MSL administration. Therefore the conclusions on the beneficial effects of MSL against metabolic syndrome was due to activated TFEB in vivo were overstated.

Specific Concerns:

1. Supplemental Figure 1e. Authors need to determine several other mTOR substrate proteins such as p-4EBP1 because it appeared #9 compound (MSL) decreased the levels of p-46K1 although it did not affect p-mTOR. Thus the conclusion that MSL was mTOR-independent autophagy inducer is weak.
2. Figure 1, all the imaging data need to be quantified. Autophagic flux assay also needs to be performed in Figure 1e because only LC3-II levels are difficult to be interpreted.

Reviewer #2 (expert in autophagy) (Remarks to the Author):

This study reports the discovery and development of a calcineurin activator that activates TFEB and autophagy. This compound ameliorates metabolic parameters on ob/ob mice and mouse models of diet-induced obesity.

The strengths of this study are aspects of the drug discovery effort and the modification of MSL to yield a better compound. However there are a number of gaps and assumptions made by the authors which are problematic and should be addressed.

1. From a conceptual perspective, the authors attribute the benefits they see to autophagy. There is a correlation but this is not necessary causal. However, they provide very little support that autophagy is the protective effector of MSL and its derivative. Calcineurin activation has many consequences besides TFEB. TFEB regulates many processes besides autophagy (possibly including aspects of metabolism). No experiments are provided to enable the conclusion that TFEB or autophagy are the causal mediators. For example, the experiments in Fig 3 would benefit by similar studies in autophagy wild-type and autophagy-null cells (for both MSL and MSL7). Ideally, such experiments could also be performed in livers where autophagy is compromised in vivo. So, labelling the effects as due to autophagy enhancement is very risky and is not supported by data

provided. The autophagy data in Supp Fig 1 should be shown more clearly for MSL.

2. The data in Fig 1e suggest that there may be some induction of autophagy by MSL in TFEB-KO cells. Is this real and, if so, then this may suggest that some effects of MSL are TFEB-independent.

3. How do the autophagy induction effects of MSL compare to other means of activating calcineurin activity (e.g. the constitutively active calcineurin mutant used in Fig 2)?

4. What are the free drug levels in the in vivo experiments for MSL and MSL-7 and how do these compare to the in vitro concentrations that have effects?

5. Is TFEB S142 phosphorylation altered by MSL and its derivative in vivo?

6. The authors should show the effects of MSL-7 on calcineurin binding and TFEB S142 phosphorylation

Reviewer #3 (expert in autophagy and metabolism)(Remarks to the Author):

In the present study, Lim et al investigate the function of a novel mTOR-independent- autophagy inducer (coined MSL) discovered through high content screen. They suggest that this compound activates TFEB through dephosphorylation by calcineurin, thus inducing lysosomal and autophagic gene expression, leading to increase in autophagic flux. To test the potential therapeutic effect, they treated animal models of diabetes with MSL and its analogue MSL-7 (which is more stable in the microsomal fraction). The treatment partially protected ob/ob mice from insulin resistance and glucose intolerance. MSL-7 was also protective of mice fed high fat diet.

While the compounds have remarkable effects in vivo and while the concept of stimulating autophagy without interfering with mTOR -in order to prevent side effects- makes sense, the study will require a more thorough approach to be convincing.

Major comments:

1) Most of the microscopy imaging and immunoblots are lacking quantification.

2) Fig.1 The quantification of autophagy is fine, but it would benefit from additional experiments to ensure the effect on autophagic flux (e.g. p62, LC3 +/- lysosome inhibitor)

3) Based on Figure 1e, the authors raise the hypothesis that MSL induces autophagy via dephosphorylation of TFEB. Yet, the concentration used to induce autophagy (50uM) did not affect TFEB phosphorylation. 100uM is the minimal concentration to have substantial dephosphorylation of TFEB. This discrepancy is also apparent in respect to calcineurin (Fig 2b).

4) Fig 2e shows that MSL protects calcineurin from pronase. Stabilization of calcineurin is therefore suggested to be the mechanism of action of MSL. I'm not however convinced with this result. The concentration of MSL in this experiment is very high (1mM) and the effect is very on calcineurin stability is not that impressive (though maybe quantification and normalization with actin would make it more convincing). Lower concentrations of MSL would be in place. Also, some negative control (MSL analogue that doesn't induce autophagy) would strengthen the belief in the result.

5) Fig. 3: The cell type is not indicated. It would be interesting to have cells that are susceptible to lipotoxicity and test whether is protective.

6) Fig 3. The effect of Orlistat suggests that the decrease in lipid droplets is not mediated by lysosomal enzymes (to my knowledge orlistat acts on cytosolic lipase but not on lysosomal lipase). To test whether autophagy is involved some autophagy inhibitors should be included (e.g. 3MA, bafilomycin).

7) Fig 3. The concentration of FFAs is rather high (400 and 800uM for palmitate and oleate respectively). It's not indicated how this mixture was prepared, and what was the ratio to BSA. Also what is the control? Is it regular medium or medium containing BSA?

8) According to the cellular model, lipid droplets degradation is at least part of the mechanism. Yet, there is no much evidence for this in vivo. Body weight is not affected by the treatment. It would be good to test whether the adipose tissue mass is reduced.

9) Figure 6: It would be good to have some more information about the effect of the MSL compound on blood metabolites and hormones (levels of lipids, insulin, C-peptide). Gluconeogenesis would also be interesting to test.

Minor comments:

10) Give more information regarding the screen. What was the positive control used? Why 0.6 was chosen as a criterion for the hits.

"... chemicals (#6, #9 and # 30) improved glucose profile of ob/ob mice after

Minor comments:

1) 82 in vivo administration for 8 weeks in our preliminary experiments (data not shown)..." Show the data in the supp.

2) Figure 1e contains a band against phosphorylated TFEB that seems unspecific.

3) FFAs inhibit the lysosomal proton pump thus reducing lysosomal acidity and activity. How activation of Tfeb by MSL would allow that is not clear to me? Address this point in the text.

4) Figure 3A: It's not clear what the first two vehicles columns are. I guess the first one is without FFAs and the second is with FFAs; please indicate.

5) Figure 3C: Indicate what are the white and black bars

6) Supplementary fig 3: I'm not convinced by the approached used for measuring mitochondrial potential and mitochondrial ROS. Use some controls and do quantification. If possible, also, measure oxygen consumption, which is the golden standard for mitochondrial function.

Reviewer #4 (expert in obesity and metabolism)(Remarks to the Author):

Overall this is an exciting publication. While the concept that autophagy can modulate insulin sensitivity and hepatic steatosis is well established, the authors present a powerful new approach to mediate it pharmacologically and show that their new compound acts independently of the mTOR pathway.

Overall my enthusiasm for the manuscript could be increased by better characterisation of the metabolic effects of the novel autophagy inhibitor. I also have a few specific points regarding the data.

While the authors show several readouts of improved glucose metabolism, the only readout addressing insulin sensitivity is the ITT. The manuscript would be strengthened with more biochemical evidence for how the changes in glucose levels and tolerance are underpinned. Insulin levels during the GTT would be informative, and fasting a fed levels of hormones such as insulin, adiponectin and in the high-fat feeding studies, leptin.

Furthermore, the authors should provide organ weights for both liver and white and brown adipose tissue. The improvement in hepatic steatosis is marked and it would be expected if the images are representative that this would manifest in reduced liver weight. Given there are no alterations in overall body weight this could represent a redistribution of fat from the liver to the adipose tissue. Further investigation of the adipose tissue should be conducted. Markers of adipogenesis (PPARy1, PPARy2, ap2) insulin sensitivity (e.g Glut4, IRS1) and lipid metabolism (Fasn, SCD1, Elovl6, DGAT1/2, LPL, ANGPTL4, HSL, ATGL) should be measured in both intraabdominal and subcutaneous white adipose tissue. The size of adipocytes should be quantified. LC3-1 and LC3-II should be measured in the leupeptin clamp experiment for adipose tissue as well as in liver (shown in figure 3e)

Given autophagy has been implicated in beige and brown adipose tissue function, measurement of brown fat markers (UCP1, Deiodinase2, PGC1a and Elovl3) should be measured in brown and subcutaneous white adipose tissue.

Specific points:

The claim that MSL alters inflammasome activation is not really supported by the data.

Consistently the effect seems to be predominantly on inflammasome priming. Supplemental 3A shows that MSL reduces ILb secretion in the presence of LPS without PA, even if it does not quite reach statistical significance. Furthermore S3b shows a huge reduction in pro IL1b. Figure 4J shows reductions in il1b mRNA expression that are nearly twice the reduction in F480, consistent with a lower per-macrophage expression of il1b. The fact Il1b and inflammasome priming is

modified is very interesting, but suggests a different, transcriptionally based mechanism that should be investigated.

With regards to figures 3A and B. The inhibitor orlistat will inhibit almost all lipases, not just LIPA. Do the HeLa cells express ATGL and HSL? Also do the residual lipid droplets following MSL treatment co-stain with Lamp1? It would be interesting if they did not and help to confirm the specificity of MSL to mediating lipid clearance from the cells by activation of autophagy.

The methods do not appear to contain a description of the leupeptin clamp experiment, unless I have missed it.

Overall this is a very good manuscript that would benefit from some more mechanistic insights into how the activation of autophagy leads to improvements in systemic glucose metabolism.

Reviewers' comments:

Reviewer #1 (expert in autophagy, apoptosis, liver injury)(Remarks to the Author):

Metabolic syndrome and diabetes are important worldwide health issues currently. In this manuscript, authors identified a novel autophagy inducer from an autophagy screening and investigated its role and underlying mechanisms against metabolic syndrome in mice. They found this compound (MSL) activated TFEB likely through activation of phosphatase calcineurin but not mTOR and in turn enhanced hepatic autophagy, which might help to remove excess hepatic lipid droplets and also improve glucose tolerance and insulin sensitivity. While the data on MSL induced beneficial effects against metabolic syndrome in ob/ob mice and diet-induced obesity mice were promising, the mechanistic link and insights on TFEB and autophagy in vivo was relatively weak. Some of the data were also not convincing.

Major Concerns:

1. Despite that MSL improved metabolic syndrome in ob/ob mice. It was unclear whether there was a defect of TFEB signaling in the ob/ob mice or high fat diet-induced obesity. Although authors provided strong data to show MSL activated TFEB in cultured cells, not data to show MSL also activated TFEB in ob/ob mice or high fat diet-treated mice. In addition, whether knockdown of TFEB in ob/ob mice would eliminate MSL beneficial effects were not determined.

Ans) We studied Tfeb in HFD-fed mice as suggested. We observed apparently increased S142-phospho-Tfeb in the liver of HFD-fed mice, suggesting compromised Tfeb activity. Furthermore, we observed markedly reduced S142-phospho-Tfeb after in vivo administration of MSL-7 to HFD-fed mice for 8 weeks, suggesting MSL activates Tfeb signaling in vivo. These data are incorporated as Supplementary Fig. 11. We also studied the effect of Tfeb knockdown on the

metabolic improvement by MSL-7 as suggested, which showed reversal of metabolic effects of MSL-7 by in vivo transfection of Tfeb siRNA. These important data were incorporated as Supplementary Fig. 12.

2. Multiple tissues including muscle, adipose and pancreatic tissues in addition to liver are all important for metabolic syndrome. However, it seemed that only the histological changes on these tissues were determined (Figure 4 e and h and supplemental figure 6) but the more important lysosomal biogenesis and autophagy activities were not determined in these tissues after MSL administration. Therefore the conclusions on the beneficial effects of MSL against metabolic syndrome was due to activated TFEB in vivo were overstated.

Ans) We studied the expression of lysosomal genes and autophagy genes in multiple tissues of mice treated with MSL-7 for 8 weeks, as suggested. In muscle, induction of lysosomal genes and autophagy genes was not observed after administration of MSL-7 for 8 wk. Instead, the expression of Mfn1, Mfn2, Nrf-1, Nrf-2, Cox1, Cox2, Tfam and citrate synthase was increased, which is consistent with a previous paper showing induction of mitochondrial biogenesis genes but not lysosomal genes by Tfeb AAV injection to muscle (Mansueto et al. Cell Metab 25:182, 2017). In white adipose tissue, the expression of lysosomal genes and Tfeb was upregulated by high-fat diet alone, consistent with a previous paper (Xu X et al., Cell Metab 18:816, 2013), which may be an adaptive changes to adapt to local lipid-rich environment. The expression of Tfeb and lysosomal genes in adipose tissue of HFD-fed mice was not further increased by MSL-7. Intriguingly, the expression of lipogenesis genes was enhanced after administration of MSL-7 for 8 wk. The expression of autophagy genes and lysosomal genes was not significantly changed in the pancreas. These data were incorporated as Supplementary Fig. 14 and 16.

Specific Concerns:

1. Supplemental Figure 1e. Authors need to determine several other mTOR substrate proteins such as p-4EBP1 because it appeared #9 compound (MSL) decreased the levels of p-46K1 although it did not affect p-mTOR. Thus the conclusion that MSL was mTOR-independent autophagy inducer is weak.

Ans) We studied the effect of MSL and MSL-7 on 4EBP1 and S6K1 phosphorylation as suggested. Phosphorylation of 4EBP1 and S6K1 was inhibited by Torin-1 but not by MSL or MSL-7, which was incorporated as Supplementary Fig. 10c.

2. Figure 1, all the imaging data need to be quantified. Autophagic flux assay also needs to be performed in Figure 1e because only LC3-II levels are difficult to be interpreted.

Ans) We quantified imaging data throughout Fig. 1 (confocal microscopy and Western blot), as suggested. Densitometric analysis of immunoblot bands was also conducted throughout the manuscript. Autophagic flux assay was conducted in the presence or absence of bafilomycin A1, which was incorporated as Fig. 1g.

Reviewer #2 (expert in autophagy) (Remarks to the Author):

This study reports the discovery and development of a calcineurin activator that activates TFEB and autophagy. This compound ameliorates metabolic parameters on ob/ob mice and mouse models of diet-induced obesity.

The strengths of this study are aspects of the drug discovery effort and the modification of MSL to yield a better compound. However there are a number of gaps and assumptions made by the authors which are problematic and should be addressed.

1. From a conceptual perspective, the authors attribute the benefits they see to autophagy. There is a correlation but this is not necessary causal. However, they provide very little support that autophagy is the protective effector of MSL and its derivative. Calcineurin activation has many consequences besides TFEB. TFEB regulates many processes besides autophagy (possibly including aspects of metabolism). No experiments are provided to enable the conclusion that TFEB or autophagy are the causal mediators. For example, the experiments in Fig 3 would benefit by similar studies in autophagy wild-type and autophagy-null cells (for both MSL and MSL7). Ideally, such experiments could also be performed in livers where autophagy is compromised in vivo. So, labelling the effects as due to autophagy enhancement is very risky and is not supported by data provided. The autophagy data in Supp Fig 1 should be shown more clearly for MSL.

Ans) To provide data supporting that autophagy enhancement is causally related to the metabolic improvement by MSL administration, we conducted in vivo knockdown of Tfeb by in vivo Tfeb siRNA transfection, which showed a significant reversal of MSL effect. This data was incorporated as Supplementary Fig. 12. Lipid clearance by MSL was studied using autophagy KO (Atg7-KO) MEF as suggested. In these cells, increased lipid clearance was abrogated in Atg7-KO cells, which was incorporated as Supplementary Fig. 5c.

In Supplementary Fig. 1, data involving MSL was shown as black bar in Supplementary Fig. 1b and red number in Supplementary Fig. 1e, as suggested.

2. The data in Fig 1e suggest that there may be some induction of autophagy by MSL in TFEB-KO cells. Is this real and, if so, then this may suggest that some effects of MSL are TFEB-independent.

Ans) In Fig. 1e, the lower band (now marked by red arrow head) appears to be a p-S142-Tfeb band because it was entirely absent in TFEB knockout cells. The

upper band might not be a p-S142-Tfeb band, since it is still observed in TFEB knockout cells. Thus, we could not see induction of the 'correct' band after MSL treatment of TFEB knockout cells in Fig. 1e. We agree with the reviewer's comment that MSL may have Tfeb-independent function since calcineurin activation can have effects other than Tfeb activation. However, Fig. 1e does not appear to support Tfeb-independent effect of MSL.

3. How do the autophagy induction effects of MSL compare to other means of activating calcineurin activity (e.g. the constitutively active calcineurin mutant used in Fig 2)?

Ans) We studied autophagy induction by transfection of constitutive active mutant of calcineurin. Transfection of constitutively active calcineurin induced autophagic activity, which was similar to that by MSL. However, direct quantitative comparison between calcineurin effect and MSL effect might be difficult because calcineurin effect can be affected by transfection efficiency. This data was incorporated as Supplementary Fig. 3c.

4. What are the free drug levels in the in vivo experiments for MSL and MSL-7 and how do these compare to the in vitro concentrations that have effects?

Ans) We measured serum level of MSL and MSL-7 by LC-MS/MS method. C_{max} of MSL-7 was 0.57 µg/ml (1.63 µM). That of MSL was 0.21 µg/ml (0.61 µM). We agree that such concentrations are much lower than that employed for in vitro experiments (50 ~ 100 µM). Thus, we conducted in vitro experiment (TFEB nuclear translocation and calcineurin activation) again using lower concentration of MSL. When we conducted intrapolation based on the titration curve in Supplementary Figure 3a,b, ca. 14.0% and 17.3% of cells would show TFEB nuclear translocation after treatment with 0.61 µM MSL or 1.63 µM MSL-7 which can be attained in

vivo. We believe such a small degree of TFEB nuclear translocation can explain metabolic effect of MSL-7 or MSL in vivo. Too much Tfeb activation for a prolonged period may have harmful effect since Tfeb family members sometimes can be oncogenic depending on tissues. These data were incorporated as Supplementary Fig. 3a,b and discussed in lines 365-370.

5. Is TFEB S142 phosphorylation altered by MSL and its derivative in vivo?

Ans) We studied in vivo phosphorylation of Tfeb after MSL-7 administration to HFD-fed mice, as suggested. TFEB S142 phosphorylation was upregulated in the liver of HFD-fed mice compared to chow-fed mice. In vivo administration of MSL-7 markedly reduced TFEB S142 phosphorylation in the liver of HFD-fed mice, which was incorporated as Supplementary Fig. 11.

6. The authors should show the effects of MSL-7 on calcineurin binding and TFEB S142 phosphorylation

Ans) We studied calcineurin binding of MSL-7 as suggested, which was incorporated as Supplementary Fig. 10b. We also studied Tfeb dephosphorylation by MSL-7, which was incorporated as Supplementary Fig. 10a.

Reviewer #3 (expert in autophagy and metabolism)(Remarks to the Author):

In the present study, Lim et al investigate the function of a novel mTOR-independent- autophagy inducer (coined MSL) discovered through high content screen. They suggest that this compound activates TFEB through dephosphorylation by calcineurin, thus inducing lysosomal and autophagic gene expression, leading to increase in autophagic flux. To test the potential therapeutic effect, they treated animal models of diabetes with MSL and its

analogue MSL-7 (which is more stable in the microsomal fraction). The treatment partially protected ob/ob mice from insulin resistance and glucose intolerance. MSL-7 was also protective of mice fed high fat diet.

While the compounds have remarkable effects in vivo and while the concept of stimulating autophagy without interfering with mTOR -in order to prevent side effects- makes sense, the study will require a more thorough approach to be convincing.

Major comments:

1) Most of the microscopy imaging and immunoblots are lacking quantification.

Ans) We quantified immunoblots by densitometric analysis in Fig.1, 2, 4, Supplementary Fig. 3, 7, 10 and 11. We also quantified imaging data, which was added to Fig. 1, 2, 3, 5, Supplementary Fig. 3, 5, and 15, as suggested.

2) Fig.1 The quantification of autophagy is fine, but it would benefit from additional experiments to ensure the effect on autophagic flux (e.g. p62, LC3 +/- lysosome inhibitor)

Ans) We determined autophagic flux employing bafilomycin A1 clamping method, which was incorporated as Fig. 1g, as suggested.

3) Based on Figure 1e, the authors raise the hypothesis that MSL induces autophagy via dephosphorylation of TFEB. Yet, the concentration used to induce autophagy (50uM) did not affect TFEB phosphorylation. 100uM is the minimal concentration to have substantial dephosphorylation of TFEB. This discrepancy is also apparent in respect to calcineurin (Fig 2b).

Ans) We agree that 100 μ M MSL significantly reduced Tfeb phosphorylation at S142. However, 50 μ M MSL also reduced Tfeb phosphorylation at S142 to some

degree. When we studied the effect of MSL-7 in the revision experiment, we observe that 50 μ M MSL-7 significantly reduced Tfeb phosphorylation at S142, which was incorporated as Supplementary Fig. 10a. When we measured in vivo concentration of MSL after administration. Cmax was 1.63 μ M for MSL-7 and 0.61 μ M for MSL. In our previous experiment, Tfeb nuclear translocation occurred in >80% of cells after treatment with 50 μ M MSL or MSL-7. In our opinion, Tfeb nuclear translocation in >80% of cells needs not happen and should not happen in vivo. We conducted the experiment (Tfeb nuclear translocation and calcineurin activation) again using lower concentration of MSL. We observed that MSL induces both TFEB nuclear translocation and calcineurin activation in a dose-dependent manner, which was incorporated as Supplementary Fig. 3a,b. When we titrated the concentration of MSL or MSL-7 inducing Tfeb nuclear translocation and conducted intrapolation, 0.61 μ M MSL and 1.63 μ M MSL-7 appear to induce Tfeb nuclear translocation in ca. 14% and 17% of cells, respectively. We believe such a small degree of Tfeb nuclear translocation and Tfeb dephosphorylation at S142 can explain metabolic effect of MSL-7 or MSL in vivo. Too much Tfeb activation for a prolonged period may have harmful effect since Tfeb family members sometimes can be oncogenic depending on tissues. These data were incorporated as Supplementary Fig. 3a,b and discussed in lines 365-370.

4) Fig 2e shows that MSL protects calcineurin from pronase. Stabilization of calcineurin is therefore suggested to be the mechanism of action of MSL. I'm not however convinced with this result. The concentration of MSL in this experiment is very high (1mM) and the effect is very on calcineurin stability is not that impressive (though maybe quantification and normalization with actin would make it more convincing). Lower concentrations of MSL would be in place. Also, some negative control (MSL analogue that doesn't induce autophagy) would

strengthen the belief in the result.

Ans) We agree with the reviewer's comment that 1 mM of MSL is a high concentration. According to the previous report of DARTS assay, initial use of 10-fold higher concentration than KD is recommended to ensure maximal protection of the target protein from proteolysis by saturating the protein with ligand (Lomenick *et al.*, *Curr Protoc Chem Biol*, 2011; Lomenick *et al.*, *Proc Natl Acad Sci*, 2009; Kim *et al.*, *J Proteome Res*, 2017). Furthermore, treating with the concentration around EC50 to highly concentrated cell lysate would not be enough in terms of titration issue, which is the reason we employed 1 mM of MSL or MSL-7.

We conducted validation assay of DARTS using 1 mM and 100 μ M of MSL for Calcineurin A. β -actin was also included as a control according to the reviewer's suggestion. As shown in our new Fig. 2e and Supplementary Fig. 4, Calcineurin A degradation was protected by a high concentration (1mM) of MSL, while it was not protected by a lower concentration (0.1 mM). These results could be explained by the requirement of a saturating dose of ligand in DARTS assay employing a highly concentrated cell extract, as discussed above. β -actin was not protected in all conditions employed. These data were incorporated as Supplementary Fig. 4 and discussed in lines 135-142.

We also conducted DARTS assay using MSL derivatives that do not enhance autophagic activity, as suggested. #9-3 or #9-4 did not protect calcineurin A even at a high concentration of 1 mM, supporting validity of our DARTS assay and the relationship between calcineurin binding vs. autophagy activation. These data were incorporated Supplementary Fig. 10b and discussed in lines 236-239.

5) Fig. 3: The cell type is not indicated. It would be interesting to have cells that are susceptible to lipotoxicity and test whether is protective.

Ans) We specified cell types in Fig. 3 legend (HeLa cells). We also studied lipotoxicity by PA and/or OA. MSL did not inhibit lipoapoptosis by PA, which was incorporated as Supplementary Fig. 5d, as suggested. Lipotoxicity experiment was conducted using Hepa1c1c hepatocytes.

6) Fig 3. The effect of Orlistat suggests that the decrease in lipid droplets is not mediated by lysosomal enzymes (to my knowledge orlistat acts on cytosolic lipase but not on lysosomal lipase). To test whether autophagy is involved some autophagy inhibitors should be included (e.g. 3MA, bafilomycin).

Ans) We agree with the reviewer's comment that orlistat is not specific for lysosomal lipase. Thus, we repeated the experiment using lalistat 2, a specific inhibitor of lysosomal lipase. We also tested the effect of bafilomycin A1 as suggested, which again reversed the lipid clearing effect of MSL. These results were incorporated as Supplementary Fig. 5b.

7) Fig 3. The concentration of FFAs is rather high (400 and 800uM for palmitate and oleate respectively). It's not indicated how this mixture was prepared, and what was the ratio to BSA. Also what is the control? Is it regular medium or medium containing BSA?

Ans) We clearly mentioned the method of FFA preparation in the Method section (lines 440-445). Control was 2% fatty acid-free BSA-DMEM prepared in the same way, which was also clearly mentioned in the Method section. The ratio of 400 uM PA to 2% BSA (FFA-free) is ca. 1.3. In the presence of OA, PA will be incorporated to lipid droplet. Thus, PA/BSA ration will be much lower.

8) According to the cellular model, lipid droplets degradation is at least part of

the mechanism. Yet, there is no much evidence for this in vivo. Body weight is not affected by the treatment. It would be good to test whether the adipose tissue mass is reduced.

Ans) As suggested, we determined fat weight after MSL treatment for 8 weeks. Epididymal fat mass was increased to a small but significant degree rather than decreased after 8 wk of MSL administration. In contrast, weight of the liver was reduced, which can explain the absence of the changes in total body weight. Increased epididymal fat weight might be due to increased adipogenesis by Tfeb, as previously reported by Salma N et al. (Mol Cell Biol 15:e00608-16, 2017), which was supported by our real-time RT-PCR analysis. These results were incorporated as Supplementary Fig. 16. The decrease of liver weight appears to be due to r increased lipophagy and reduced triglyceride content (Fig. 4e,f)

9) Figure 6: It would be good to have some more information about the effect of the compound on blood metabolites and hormones (levels of lipids, insulin, C-peptide). Gluconeogenesis would also be interesting to test.

Ans) Serum level of lipids, insulin, C-peptide and other metabolic hormones were determined as suggested, which was incorporated as Fig. 6h-m. We also studied the expression of gluconeogenesis genes, as suggested. The expression of glucose-6-phosphatase, PEPCK, fructose 1,6 bisphosphatase or pyruvate carboxylase which was increased in the liver of mice fed HFD, was reduced by MSL-7, probably due to improved insulin signaling. These results were incorporated as Supplementary Fig. 13.

10) Give more information regarding the screen. What was the positive control used? Why 0.6 was chosen as a criterion for the hits.

Ans) Positive control was rapamycin, a well-known autophagy enhancer. 0.6 was chosen since 250 nM rapamycin reduced Renilla-LC3 luciferase fluorescence ratio to 0.6. This was clearly explained in lines 74-76.

Minor comments:

1) "... chemicals (#6, #9 and # 30) improved glucose profile of ob/ob mice after in vivo administration for 8 weeks in our preliminary experiments (data not shown) ..." Show the data in the supp.

Ans) We showed the data without structural information as Supplementary Fig. 2, as suggested.

2) Figure 1e contains a band against phosphorylated TFEB that seems unspecific.

Ans) We agree with the reviewer's comment that the upper band of immunoblot using anti-p-S142-Tfeb is nonspecific. However, one band (lower one) seems to be specific since the band was not seen when Tfeb KO cells were employed. We clearly indicated the specific bands with red arrow head in the revised paper (Fig. 1e).

3) FFAs inhibit the lysosomal proton pump thus reducing lysosomal acidity and activity. How activation of Tfeb by MSL would allow that is not clear to me? Address this point in the text.

Ans) We agree with the reviewer's comment that FFA affects lysosomal pH. In Fig. 3, we studied clearance of lipid droplet formed by loading of PA in combination with OA. Thus, target of lipid clearance is lipid droplet (containing triglyceride as the most abundant lipid) rather than FFA. PA alone does not induce formation of lipid droplet. We observed that MSL does not inhibit lipotoxicity by FFA (see our

answer to the inquiry No. 5). This point was incorporated as Supplementary Fig. 5d and discussed in lines 160-165, as suggested.

4) Figure 3A: It's not clear what the first two vehicles columns are. I guess the first one is without FFAs and the second is with FFAs; please indicate.

Ans) (-) was solvent only (2% fatty acid-free BSA-DMEM), and the second column was 400 uM PA+800 uM OA in 2% fatty acid-free BSA-DMEM. We indicated the condition of vehicle treatment in the legend of Fig.3 (lines 807-810) and also in the Method section (440-445), as suggested.

5) Figure 3C: Indicate what are the white and black bars

Ans) In Fig. 3C, color of the bars does not mean anything. We feel sorry for the confusion. We changed the color of the bars to black in Fig. 3C to avoid confusion.

6) Supplementary fig 3: I'm not convinced by the approached used for measuring mitochondrial potential and mitochondrial ROS. Use some controls and do quantification. If possible, also, measure oxygen consumption, which is the golden standard for mitochondrial function.

Ans) Mitochondrial ROS and potential changes were quantified together with positive (rotenone) and negative controls, as suggested. We also measured mitochondrial oxygen consumption using Seahorse analyzer. These data were incorporated as Supplementary Fig. 6c-e.

Reviewer #4 (expert in obesity and metabolism)(Remarks to the Author):

Overall this is an exciting publication. While the concept that autophagy can modulate insulin sensitivity and hepatic steatosis is well established, the authors present a powerful new approach to mediate it pharmacologically and show that their new compound acts independently of the mTOR pathway.

Overall my enthusiasm for the manuscript could be increased by better characterisation of the metabolic effects of the novel autophagy inhibitor. I also have a few specific points regarding the data.

While the authors show several readouts of improved glucose metabolism, the only readout addressing insulin sensitivity is the ITT. The manuscript would be strengthened with more biochemical evidence for how the changes in glucose levels and tolerance are underpinned. Insulin levels during the GTT would be informative, and fasting and fed levels of hormones such as insulin, adiponectin and in the high-fat feeding studies, leptin.

Furthermore, the authors should provide organ weights for both liver and white and brown adipose tissue. The improvement in hepatic steatosis is marked and it would be expected if the images are representative that this would manifest in reduced liver weight. Given there are no alterations in overall body weight this could represent a redistribution of fat from the liver to the adipose tissue.

Further investigation of the adipose tissue should be conducted. Markers of adipogenesis (PPAR α 1, PPAR α 2, ap2) insulin sensitivity (e.g. Glut4, IRS1) and lipid metabolism (Fasn, SCD1, Elovl6, DGAT1/2, LPL, ANGPTL4, HSL, ATGL) should be measured in both intraabdominal and subcutaneous white adipose tissue. The size of adipocytes should be quantified. LC3-I and LC3-II should be measured in the leucine clamp experiment for adipose tissue as well as in liver (shown in figure 3e)

Given autophagy has been implicated in beige and brown adipose tissue function, measurement of brown fat markers (UCP1, Deiodinase2, PGC1 α and Elovl3) should be measured in brown and subcutaneous white adipose tissue.

Ans) We measured serum levels of insulin, C-peptide, adiponectin, leptin and calculated HOMA-IR index as suggested, which were incorporated as Fig. 6h-m. Weight of the liver and adipose tissues were also measured. Liver weight was reduced by MSL-7 administration in vivo, as predicted by the reviewer. Weight of epididymal white adipose tissue (eWAT) was increased, again as predicted by the reviewer. Thus, we also studied the expression of adipogenesis genes in eWAT, as suggested. The expression of PPAR γ 2, SCD1, C/EBP α , C/EBP β or Fasn was increased after MSL-7 treatment in vivo, which may be able to explain increased eWAT weight. The expression of Angptl4 was increased probably as a reactive change to increased lipogenesis gene expression. The expression of IRS1 and Glut4 was not changed. These data were incorporated as Supplementary Fig. 16d,e. The size of adipocytes was also measured, which was incorporated as Supplementary Fig. 16f.

We tried to clamp lysosomal steps of autophagy in adipose tissue after in vivo leupeptin administration. However, we found that LC3-II in adipose tissue is not increased by leupeptin administration, which is in contrast to the liver and probably due to difficulty of leupeptin to reach adipose tissue after tail vein injection (data not shown). We thus employed CAG-RFP-GFP-LC3-transgenic mice and administered MSL-7 to study autophagic activity in adipose tissue without leupeptin administration. The number of RFP puncta was increased after MSL-7 administration in vivo, showing that MSL-7 exerted its activity on adipose tissue. These results were incorporated as Supplementary Fig. 15. We also studied expression of UCP1, Deiodinase2, PGC1a and Elovl3 in BAT and subcutaneous fat, which was incorporated as Supplementary Fig. 16b,c.

Specific points:

The claim that MSL alters inflammasome activation is not really supported by the

data. Consistently the effect seems to be predominantly on inflammasome priming. Supplemental 3A shows that MSL reduces IL1b secretion in the presence of LPS without PA, even if it does not quite reach statistical significance. Furthermore S3b shows a huge reduction in pro IL1b. Figure 4J shows reductions in il1b mRNA expression that are nearly twice the reduction in F480, consistent with a lower per-macrophage expression of il1b. The fact IL1b and inflammasome priming is modified is very interesting, but suggests a different, transcriptionally based mechanism that should be investigated.

With regards to figures 3A and B. The inhibitor orlistat will inhibit almost all lipases, not just LIPA. Do the HeLa cells express ATGL and HSL? Also do the residual lipid droplets following MSL treatment co-stain with Lamp1? It would be interesting if they did not and help to confirm the specificity of MSL to mediating lipid clearance from the cells by activation of autophagy.

The methods do not appear to contain a description of the leupeptin clamp experiment, unless I have missed it.

Ans) We agree with the reviewer's comment that proIL-1b level was reduced by MSL, which could be an autophagy-independent event. Thus, we determined mRNA level of cytokines such as pro-IL1 β , TNF α and IL-6. Indeed, mRNA expression of those cytokines was significantly reduced by MSL, supporting transcription-dependent mechanisms. We investigated the mechanism of such findings, as suggested. We found that NF- κ B activation was reduced by MSL, which is consistent with previous papers showing that calcineurin activation inhibits NF- κ B signaling through TLR4, MyD88 or TRIF binding (Kang YJ et al. J Immunol 179:4598, 2007; Conboy I et al. PNAS 96:6324, 1999). Specifically, I κ B α phosphorylation, disappearance of I κ B α , p65 phosphorylation induced by LPS was attenuated by MSL or MSL-7. Furthermore, LPS-induced NF- κ B reporter activity measured using pELAM-luciferase NF- κ B reporter construct was also attenuated by MSL or MSL-7. This autophagy-independent mechanism can contribute to the

beneficial effect of MSL on metabolic inflammation and glucose profile in addition to the autophagy-dependent mechanism, the main mechanism of the therapeutic effect of MSL. This important data was incorporated as Supplementary Fig. 7, and we thank reviewer for critical comments.

We also agree with the reviewer's comment that orlistat is not specific for lysosomal lipase. Thus, we conducted the experiment again using lalistat 2, a specific inhibitor of lysosomal lipase. This data was incorporated as Supplementary Fig. 5b. We observed that HeLa cells express ATGL and HSL (data not shown). We observed no colocalization between LAMP-1 and BODIPY in the residual fat droplet except a few droplets after 24 h of MSL treatment, while we observed extensive colocalization after 1 h of MSL treatment. These data were incorporated as Supplementary Fig. 5a.

We detailed the procedure of leupeptin experiment in the Method section, as suggested (lines 527-528).

Overall this is a very good manuscript that would benefit from some more mechanistic insights into how the activation of autophagy leads to improvements in systemic glucose metabolism.

REVIEWERS' COMMENTS:

Reviewer #1 (Remarks to the Author):

Authors have addressed my concerns. I am satisfied with the revision.

Reviewer #2 (Remarks to the Author):

I am satisfied with the responses to the reviewers' comments and the new experiments

Reviewer #4 (Remarks to the Author):

The authors have done a great job of addressing my questions and they have significantly strengthened the manuscript.